# The Road Less Traveled: Enhancing Exploration in LLMs via Sequential Sampling

*Two roads diverged in a wood, and I—*
*I took the one less traveled by,*
*And that has made all the difference.*

—Robert Frost, *The Road Not Taken*

## Abstract

Reinforcement learning (RL) has been pivotal in enhancing the reasoning capabilities of large language models (LLMs), but it often suffers from limited exploration and entropy collapse, where models exploit a narrow set of solutions, leading to a loss of sampling diversity and subsequently preventing RL from further improving performance. This issue is exacerbated in parallel sampling methods, where multiple outputs are drawn from the same distribution, potentially causing the model to converge to similar solutions. We propose **SESA**, a novel **SE**quential **SA**mpling framework that mitigates this challenge by generating diverse solution sketches sequentially before expanding them into full reasoning paths. This approach ensures broader exploration by conditioning each new output on previous ones, promoting diversity throughout the process and preventing policy collapse. Our experiments on a synthetic task show that sequential sampling consistently outperforms traditional RL methods in terms of path diversity and recovery from collapse. Further evaluations on real-world tasks demonstrate that SESA improves both the exploration of valid strategies and the overall performance of LLMs. On three agent benchmarks, SESA lifts success rates by $+0.25$, $+0.42$, and $+0.07$ absolute over the base model (up to an additional $211\%$ relative improvement over baseline RL), underscoring its exploration advantage. This work introduces a structured approach to exploration, paving the way for more effective and diverse reasoning in RL-trained LLMs. Code can be found at https://anonymous.4open.science/r/SESA-5E63.

## 1 Introduction

Reinforcement learning with verifiable rewards (RLVR) has become a cornerstone for enhancing the reasoning capabilities of large language models (LLMs). RLVR enables models to generate multiple solutions to a given problem, using a reward function that incentivizes correct answers or penalizes incorrect ones, based on verifiable criteria. This approach has demonstrated significant improvements in LLM performance, such as achieving higher one-shot accuracy, producing longer chains of reasoning, and enhancing self-checking behaviors (Gao et al., 2024; Lambert et al., 2025; DeepSeek-AI et al., 2025; Hu et al., 2025). Consequently, RLVR has emerged as a crucial method for scaling the reasoning abilities of LLMs in complex tasks.

A fundamental principle of reinforcement learning is that **exploration drives continual improvement**. Models need to balance exploiting known high-value strategies with exploring uncertain areas of the state-action space to uncover new information and diverse solutions (Sutton et al., 1998). However, in the context of LLMs, current RL paradigms often struggle to achieve this balance, leading to a preference for conservative actions and premature convergence to local optima (Yue et al., 2025). As RL training repeatedly shift probability mass toward already high-reward outputs, exploration becomes increasingly suppressed, narrowing the space of plausible solution strategies. This is particularly problematic when using pass@k evaluations, where a base model's output diversity

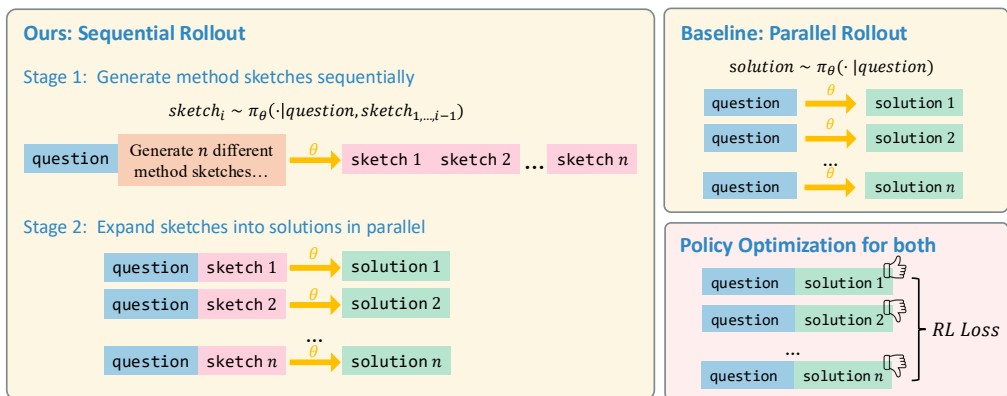

Figure 1: Training overview. Baseline parallel rollout (left) samples all solutions i.i.d. from the same distribution, while our sequential rollout (right) first generates diverse methods sequentially, then expands each into full solutions in parallel.

can sometimes exceed that of an RL-trained model, despite the latter achieving higher individual performance metrics. This contraction of the "reasoning frontier" hampers the discovery of novel strategies, ultimately limiting the compounding improvement that RL seeks to achieve.

In the face of this challenge, various methods have been proposed to encourage exploration through **entropy regularization**. By incorporating entropy-based rewards or adjusting the sampling strategy to introduce more randomness (Vanlioglu, 2025; Cui et al., 2025), these techniques aim to preserve diversity in the model's output while still optimizing for correctness. While effective to some extent, these methods rely heavily on parallel sampling paradigms, such as those used in algorithms like GRPO (Shao et al., 2024) and its variants (Yu et al., 2025; Liu et al., 2025a; Zheng et al., 2025a). In these approaches, multiple completions are generated independently for the same input, we refer to this method of generation as *parallel sampling*. Since each sample is drawn independently and identically from the same policy, it is prone to generating similar outputs. As a consequence, as the RL policy converges to a high-reward solution, the outputs become **increasingly similar** across different samples, causing the model to lose diversity and effectively **"stall" in its learning**. Once this policy collapse occurs, training becomes **ineffective** because the model is **unable to explore new strategies**. Some work has introduced tree-search or MCTS into rollout stage to obtain dense reward signals (Hou et al., 2025; Guo et al., 2025). However, these methods still face the issue where the different child nodes at each step cannot see each other, and thus cannot guarantee the generation of diverse methods or ensure diversity. Similarly, they may also encounter policy collapse, which can hinder effective learning by preventing the model from exploring new strategies.

To address this challenge, we propose a shift in the sampling paradigm by using *sequential sampling*. As shown in Figure 1, Unlike parallel sampling, where multiple completions are generated independently from the same policy, sequential sampling generates solutions one by one, **conditioning each new output on the previous ones**. This approach can actively steer the model away from previously generated solutions, ensuring that each new candidate is sufficiently **distinct from its predecessors**, leading to significantly enhanced sampling diversity as shown in Figure 2 (see Section 2.2 for details). By progressively diversifying the generated solutions, sequential sampling fosters **greater exploration** and improves the model's ability to **discover a broader range of valid strategies**.

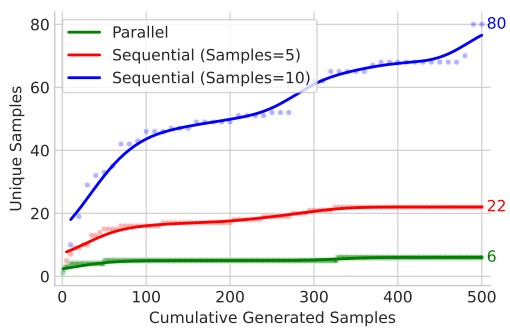

Figure 2: Parallel sampling quickly plateaus at 11 species, while sequential sampling continues to discover new species, reaching 22 and 80, respectively.

Given the complexity of real-world tasks and the challenge of managing long outputs, we introduce a two-stage procedure to maintain both diversity and efficiency in complex reasoning tasks. In this approach, we first sequentially generate concise "method sketches" that outline distinct strategies, and then generate full solutions based on these different sketches. This two-stage framework preserves sample diversity while maintaining computational efficiency, providing a robust solution for real-world RL applications.

In the experiments, we first demonstrate the effectiveness of sequential sampling in a controlled synthetic task, where we show that it: (1) uncovers strategies that parallel sampling fails to discover, and (2) retains a larger proportion of correct solutions compared to traditional RL approaches. To further evaluate its potential, we test sequential sampling on five different tasks incorporating agent, reasoning, and math, each of which requires extensive exploration in order to achieve meaningful improvements. Our sequential sampling method consistently outperforms strong, established parallel sampling baselines such as DAPO and RAGEN.

Our primary contributions are as follows:

- **Sequential Sampling Framework**: We propose a two-stage framework SESA for SEquential SAmpling in reinforcement learning that overcomes the limitations of traditional parallel sampling. By conditioning each new output on previously generated solutions, sequential sampling promotes greater exploration and diversity in the generated paths.
- **Diversity Enhances RL Training**: Our experiments show that increasing output diversity during RL training leads to continuous performance improvements by maintaining better exploration.
- **Comprehensive Analysis**: We provide a detailed analysis of the experimental results, showing that sequential sampling produces a wider range of solutions that parallel sampling fails to capture. SESA enhances solution diversity during RL training and can even revive collapsed strategies, restoring the model's capacity to discover new valid paths and reigniting the training process.

## 2 MOTIVATING TASK: SYNTHETIC PATH EXPLORATION

### 2.1 TASK DESCRIPTION

To systematically evaluate whether sequential sampling can expand a model's ability to discover a diverse set of valid strategies, we introduce a controlled synthetic task—*Path Exploration*. In this task, a fixed list of 20 valid location strings (formatted as Province–City–District/County, e.g., Guangdong-Shenzhen-Nanshan) is prepared, each representing a hidden treasure point in China. The model is asked to generate one possible treasure point at a time, and if the generated point matches one of the 20 predefined ones, it receives a reward of 1; otherwise, it receives a reward of 0.

This task is designed to simulate real-world problems that require exploration across a wide range of possible solutions. Many real-world problems admit *multiple correct solutions or workflows* (e.g., diverse reasoning chains, different game trajectories). A model trained with RL often collapses onto a few high-reward patterns, discarding other equally valid paths. *Path Exploration* serves as a reliable proxy for this challenge: each treasure point corresponds to one "correct path", and discovering many distinct treasures corresponds to retaining a broader set of valid strategies. Our central question is whether sequential sampling can help RL-trained policies *retain more correct paths* instead of converging to a narrow subset. A detailed description of the data curation process is in A.1.

We quantify exploration breadth and output variation using metric **Coverage@$k$**: Let the model output $k$ candidates for an instance. If $u$ distinct treasures are hit among those candidates and the total number of treasures is $U=20$, then

$$\text{Coverage@}k \;=\; \frac{u}{U}.$$

Higher Coverage@$k$ indicates that the policy preserves more distinct correct paths.

### 2.2 SEQUENTIAL SAMPLING BOOSTS DIVERSITY

In RL algorithms like GRPO, models typically generate multiple outputs (or candidates) for a given input by drawing them independently from the same probability distribution. We call it *parallel sampling*. For example, given an input $x$, we would generate $G$ independent candidates $y^{(i)}$ for

$i = 1, \ldots, G$ using the model's autoregressive policy $\pi_\theta$, like $y^{(i)} \sim \pi_\theta(\cdot \mid x)$, $i = 1, \ldots, G$. However, in our *sequential sampling*, we prompt the model to generate $G$ different paths in an autoregressive way, i.e., $y^{(i)} \sim \pi_\theta(\cdot \mid x, y^{(1)}, \cdots, y^{(i-1)})$, $i = 1, \ldots, G$. As new generated paths can see previous ones, this ensures that each path is different from the others.

Figures 2 and 3 present the results. To summarize the key findings of our experiments, we highlight the following conclusions about the advantages of sequential sampling:

***Conclusion I.*** *Sequential sampling uncovers strategies not discovered by parallel sampling.*

To illustrate how sequential sampling offers greater diversity, we conducted a simple path generation experiment. In each query, we used Deepseek-V3.1 API to generate the aforementioned valid paths, setting the temperature to 1, which is a common setting in RL training. In each API call, parallel sampling produced only one path, while sequential sampling produced 5 or 10 distinct paths. We queried the model multiple times to obtain a large number of paths. To ensure a fair comparison, we fixed the total number of generated paths to 500 across all regimes and then counted the number of distinct paths obtained. Specifically, the *parallel* setting calls 500 API times, producing 1 path per call; the *sequential-5* setting calls api 100 times producing 5 paths per call; and the *sequential-10* setting calls api 50 times producing 10 paths per call. As shown in Figure 2, parallel sampling eventually covered only 11 distinct paths and quickly plateaued. As the number of samples increased, it became increasingly unlikely for parallel sampling to discover new paths. In contrast, sequential sampling displayed stronger fluctuations, enabling it to continue uncovering previously unseen paths and ultimately reaching 22 and 80 distinct paths, respectively, with an ongoing upward trend.

***Conclusion II.*** *RL with sequential sampling retains a larger proportion of correct outputs.*

We further incorporate sequential sampling into RL training, where each rollout generates 16 distinct samples within a single response. The Qwen2.5-7B-Instruct (Qwen et al., 2025) model is trained with RL based on the DAPO algorithm (parallel sampling with $G = 16$), which serves as the baseline for comparison in this experiment. By contrast, SESA uses sequential sampling, producing 16 distinct candidates sequentially within a single response. As mentioned earlier, we aim to achieve a higher Coverage@32, indicating that the model explores and retains more correct outputs. As shown in Figure 3, in the initial steps, the model's coverage is 0, as it has not generated any correct answers. At the 3rd evaluation step, in parallel sampling, once the DAPO samples a correct answer, the gradient updates focus on **reinforcing that answer**. The probability of

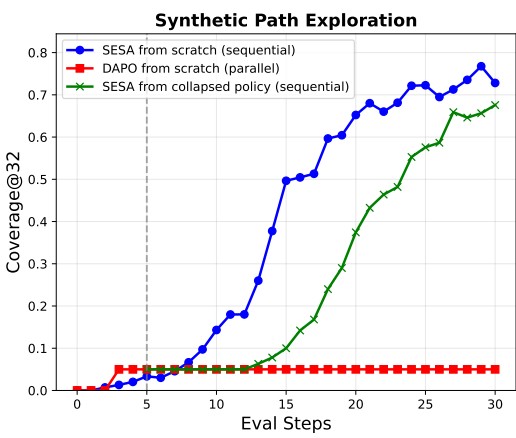

Figure 3: Synthetic path exploration result.

sampling this answer increases for all rollouts, leading to all outputs converging to this single path, causing the coverage to stabilize at $1/20 = 0.05$. However, by using sequential sampling, SESA samples 16 distinct outputs at each step, which actively diversifies the outputs, ensuring that it does not collapse into a single path. Although the coverage increases slowly in the beginning, it consistently rises to 77%, reflecting the method's ability to retain a wider range of correct paths and explore further during the training process.

***Conclusion III.*** *Sequential sampling allows models that have collapsed into very limited outputs to recover diversity.*

As analyzed earlier, after the 5th evaluation step, DAPO has exhausted its exploration and continues to output a single, repetitive solution. At this stage, there are no gradients left for RL to improve the model's performance, meaning the model essentially "stalls" in its learning process. To overcome this issue, we resumed training from the 5th evaluation step checkpoint, but with the application of sequential sampling. Unlike the parallel sampling approach, which repeatedly reinforces the same pattern, sequential sampling forces the model to generate different outputs at each step. As illustrated

---

**Algorithm 1** RL with Two-Stage Sequential Sampling

---

1: **Input:** dataset $\mathcal{D}$, policy $\pi_\theta$, batch size $B$, number of solutions $G$, reward $R$
2: **for** iteration $= 1, 2, \ldots$ **do**
3:     $\mathcal{L} \leftarrow 0$
4:     Sample mini-batch $\mathcal{B}_x \subset \mathcal{D}$ with $|\mathcal{B}_x| = B$
5:     **for** each $x \in \mathcal{B}_x$ **do**
6:         **Stage I: Sequential method drafting**
7:         $\mathcal{S} \leftarrow \varnothing$                                             ▷ history of sketches
8:         **for** $j = 1$ to $G$ **do**
9:             $p_{\text{sk}} \leftarrow \text{PromptSketch}(x, \mathcal{S})$
10:          $s_j \sim \pi_\theta(\cdot \mid p_{\text{sk}})$
11:          $\mathcal{S} \leftarrow \mathcal{S} \cup \{s_j\}$
12:         **end for**
13:         **Stage II: Guided solution generation**
14:         $\mathcal{Y} \leftarrow \varnothing$                                         ▷ all candidates for this $x$
15:         **for** $j = 1$ to $G$ **do**                ▷ solutions are expanded in parallel
16:           $p_{\text{sol}} \leftarrow \text{PromptSolve}(x, s_j)$
17:           $y_j \sim \pi_\theta(\cdot \mid p_{\text{sol}})$
18:           $r_j \leftarrow R(x, y_j)$
19:           $\mathcal{Y} \leftarrow \mathcal{Y} \cup \{(j, y_j, r_j)\}$
20:         **end for**
21:         Compute advantages $\{A_j\}$ from $\{r_j\}$
22:     **end for**
23:     Compute $\mathcal{L}$ using Equation 1.
24:     Update $\theta \leftarrow \theta - \eta \nabla_\theta \mathcal{L}$
25: **end for**

---

by the green line in Figure 3, sequential sampling effectively revitalized the model's exploration. As the training continued, the model's coverage gradually increased, reaching higher levels of diversity.

## 3 METHOD

In this section, we formally present our SESA (Sequential Sampling) RL framework.

### 3.1 TRAINING STAGE

The fully-sequential rollout used in our synthetic *Path Exploration* task maximizes inter-sample diversity by conditioning each candidate on the entire history, and generating all complete solutions in a single pass. While this is ideal for short responses, directly sequentializing $m$ full solutions in complex real-world tasks (e.g., math or code problems with long CoT) raises two practical issues: **(i) Efficiency & context budget.** Compared with parallel decoding, a fully sequential response stream can become excessively long, potentially exceeding the model's context length and causing high generation latency. **(ii) Instruction drift under long contexts.** As the history grows, the base model may fail to keep candidates self-consistent: later solutions tend to reuse intermediate steps from earlier ones, so the "solutions" are no longer complete, self-contained trajectories.

To address these constraints, we adopt a *two-stage sequential sampling* procedure that keeps the benefits of history-aware diversification while bounding latency and context length. The full algorithm is presented in Algorithm 1.

**Stage I: Sequential method drafting**   Given an input $x$, we first generate $m$ different concise *method sketches* $\{s_j\}_{j=1}^m$ *sequentially*. Each sketch is a brief plan (e.g., a search strategy or algorithmic route), explicitly designed to differ from the sketches generated earlier:

$$s_j \sim \pi_\theta\big(\cdot \,\big|\, x,\, \mathcal{S}_{j-1}\big), \quad \mathcal{S}_{j-1} = \{s_1, \ldots, s_{j-1}\}.$$

Because sketches are short, generating them sequentially adds little latency and does not stress the context window.

| Method | Sokoban | | Countdown | | FrozenLake | |
|---|---|---|---|---|---|---|
| | Succ. Rate | Δ vs Base | Succ. Rate | Δ vs Base | Succ. Rate | Δ vs Base |
| Base Model | 0.09 | — | 0.15 | — | 0.19 | — |
| RAGEN | 0.16 | +0.07 / +77.8% | 0.50 | +0.35 / +233.3% | 0.21 | +0.02 / +10.5% |
| RAGEN+Entropy | 0.15 | +0.06 / +66.7% | 0.53 | +0.38 / +253.3% | 0.21 | +0.02 / +10.5% |
| **SESA** | **0.34** | **+0.25 / +277.8%** | **0.57** | **+0.42 / +280.0%** | **0.26** | **+0.07 / +36.8%** |

Table 1: Success rates on three RL agent benchmarks. $\Delta$ indicates the absolute improvement and the relative percentage gain compared to the Base Model. Best results are highlighted in bold.

**Stage II: Guided solution generation** Next, we expand each sketch into a full solution *in parallel*, conditioning only on $x$ and the corresponding sketch:

$$y_j \sim \pi_\theta\big( \cdot \,\big|\, x,\ s_j\big), \quad j = 1, \cdots, G.$$

Importantly, the expansion of each sketch can be executed in parallel. This parallel expansion restores throughput while preserving uniqueness and self-consistency: every solution is anchored to a distinct plan and cannot borrow intermediate results from other solutions.

**Update Policy** While candidates are generated sequentially to promote diversity, the training objective evaluates each candidate only under the original input $x$. This helps maintain consistency between training and evaluation, preventing a shift in the input distribution. Specifically, for a fixed $x$ with sequentially produced candidates $\{y_i\}_{i=1}^G$ and rewards $R_i = \text{Reward}(x, y_i)$, we compute advantages as introduced in GRPO (and later adopted by DAPO), computed from $\pi_\theta(y_i|x)$ without referencing any history in the likelihood term. The objective is

$$\mathcal{J}(\theta) = \mathbb{E}_{x \sim \mathcal{D}, \{y_i\}_{i=1}^G \sim \pi_{\theta_{\text{old}}}(\cdot|x)}$$
$$\left[ \frac{1}{G} \sum_{i=1}^G \frac{1}{|y_i|} \sum_{t=1}^{|y_i|} \Big( \min\Big( r_{i,t}(\theta)\hat{A}_{i,t},\ \text{clip}\left( r_{i,t}(\theta), 1 - \varepsilon, 1 + \varepsilon \right) \hat{A}_{i,t} \Big) \Big) \right], \tag{1}$$

where $r_{i,t}(\theta) = \frac{\pi_\theta(y_{i,t}|x, y_{i,<t})}{\pi_{\theta_{\text{old}}}(y_{i,t}|x, y_{i,<t})}$. and $\hat{A}_{i,t} = \frac{R_i - \max(\{R_i\}_{i=1}^G)}{\text{std}(\{R_i\}_{i=1}^G)}$.

**Sequential Sampling for Agent** For agent-style decision making, we apply sequential sampling at the *action* level. At each decision step, the policy proposes up to $G$ distinct actions sequentially, each conditioned on what has already been proposed to discourage duplicates. Unrolling this for $d$ steps gives a branching search that can produce up to $m^d$ trajectories in the worst case, though in practice the number is smaller due to invalid moves and early terminations. To keep compute bounded, we set a global cap $U$ on the total number of expanded trajectories and take the minimum between the generated count and $U$. This preserves diversity at each step while keeping rollout cost tractable.

## 3.2 Evaluation Stage

After training, inference proceeds exactly as with a standard RL-trained LLM. Given a test input $x$, the model decodes a *single* response $\hat{y} \sim \pi_\theta(\cdot \mid x)$. No additional conditioning on prior candidates is used at evaluation time. Metrics are computed directly on $\hat{y}$, matching conventional evaluation for standard LLMs.

## 4 Sequential Sampling Benefits RL Training

We evaluate sequential sampling across diverse settings—symbolic sudoku (Radcliffe), mathematical reasoning (AIME24 (Veeraboina, 2024)), and three classic agent environments (Sokoban, Countdown, FrozenLake following Wang et al. (2025))—and show that integrating it into RL improves policy diversity, enabling sustained performance gains.

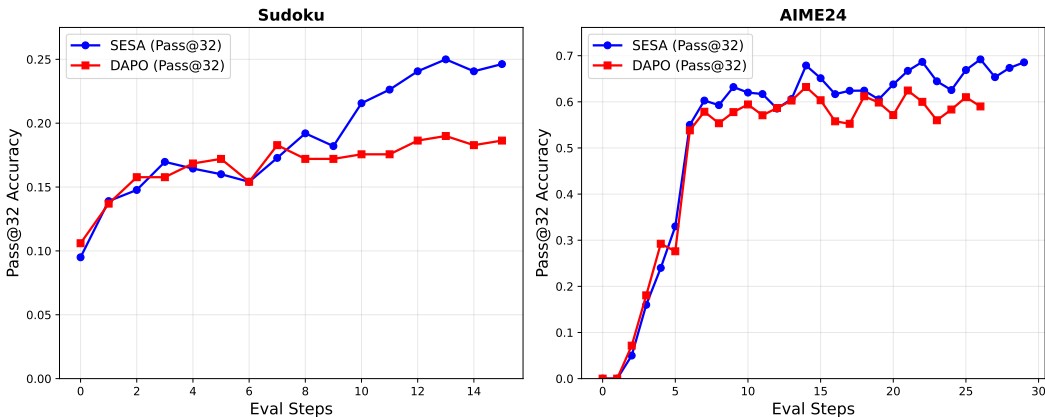

Figure 4: Performance comparison on Sudoku and AIME24. The results show the effectiveness of our two-stage sequential sampling method in improving solution diversity, particularly in higher Pass@k metrics.

## 4.1 AGENT RESULTS

Effective exploration is crucial in agent tasks: without sufficient search over possible actions and strategies, the agent easily converges to suboptimal policies. Especially in environments with sparse or deceptive rewards, sustained exploration determines whether the agent can discover diverse strategies and gradually improve its success rate.

we selected three classic reinforcement learning tasks: Sokoban, Countdown, and FrozenLake, which are commonly used to evaluate an agent's decision-making capabilities. Detailed descriptions of these tasks can be found in Appendix A.2. These tasks were chosen to simulate more challenging scenarios. To increase the difficulty, we deliberately chose a smaller model with a initial success rate of less than 20%. Given this situation, the agent must explore new strategies that go beyond the approaches chosen by the base model in order to solve more problems. We use a popular Agent framework RAGEN (Wang et al., 2025) and its variant with entropy regularization as the baselines, and we implement the sequential sampling on top of RAGEN.

In Table 1, we present the results of different methods on the three tasks. The results suggest that both RAGEN and the version with entropy regularization show only relatively modest improvements on 2 out of 3 tasks over the baseline model. For instance, in the Sokoban task, RAGEN performs nearly identically to the baseline, with the entropy-regularized version showing a simillar improvement of 0.06. Similarly, in the FrozenLake task, RAGEN's improvement is just 0.02, indicating limited effectiveness.

In contrast, SESA demonstrates substantial improvements across all tasks. Notably, on Sokoban, SESA improves by 0.25 over the base model—a 211% larger improvement than RAGEN with entropy regularization—demonstrating a significant boost. In the FrozenLake and Countdown tasks, we observe improvements of 0.07 and 0.42, respectively. These results highlight that, compared to RAGEN and the entropy-regularized version, our approach offers a much larger relative gain, particularly in terms of preserving diversity and enhancing exploration during training.

## 4.2 GENERAL TASK

To validate our proposed two-stage sequential sampling method, we conducte experiments on the Sudoku and math tasks. The results are shown in Figure 4. In the Sudoku task, we train Qwen2.5-7B-Instruct and use the Kaggle Sudoku dataset for training and testing. In the first stage, we sequentially sample the directions for filling the Sudoku grid, ensuring diversity in the generated steps. In the second stage, the model complete the entire Sudoku puzzle. Our method improved the success rate by 6% compared to the RAGEN baseline, demonstrating stronger exploration capabilities.

For the math task, we train Qwen2.5-7B-Instruct on the DAPO-17k-math dataset (Yu et al., 2025) and test using the AIME24 set. In the first stage, the model generate the overall strategy for solving

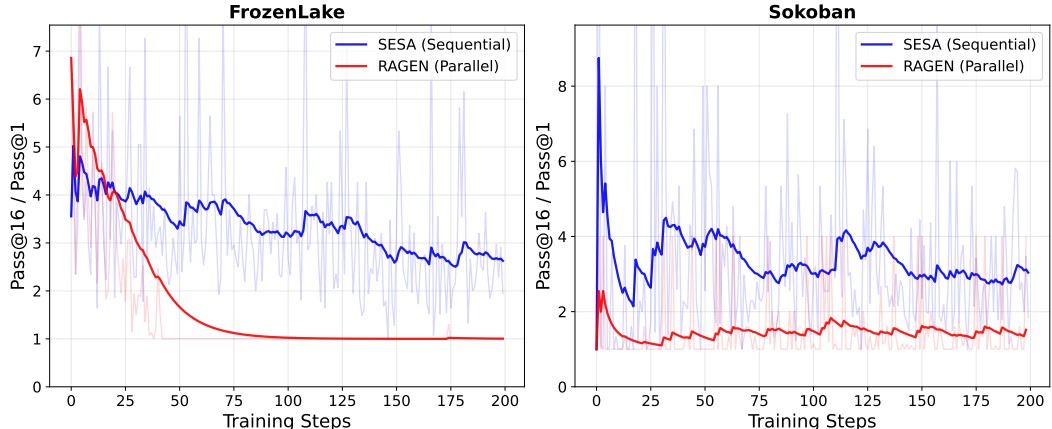

Figure 5: Comparison of Pass@16/Pass@1 ratios for sequential and parallel sampling. Sequential sampling maintains higher diversity, while parallel sampling declines, indicating a collapse into a "dead policy" with reduced exploration.

the problem, and in the second stage, it complete the solution process. Our two-stage approach achieves comparable performance to the baseline in Pass@1, but improved Pass@k by 9%. This indicates that our model retains more diverse outputs.

# 5   HOW SAMPLING DIVERSITY BENEFITS RL TRAINING

In our experiments, we observed that sequential sampling significantly outperforms parallel sampling in terms of maintaining solution diversity. Specifically, we found that SESA achieved a higher ratio of Pass@16/Pass@1, as shown in Figure 5. This indicates better diversity in the solutions, as the model is able to solve the problem after more attempts, even if it doesn't succeed on the first try. This leaves the model with the potential to further improve its performance via RL.

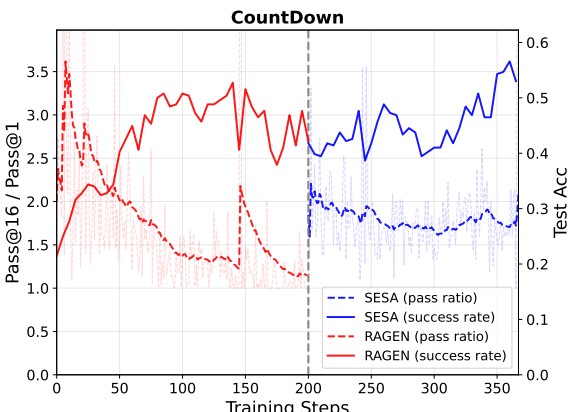

Figure 6: SESA can recover from collapsed policy.

In contrast, the Pass@16/Pass@1 ratio for the parallel sampling method gradually decreased over training. This trend was especially evident in tasks like FrozenLake and Countdown, where the ratio dropped to 1, indicating that the model had collapsed into a "dead policy." In this state, the model's outputs became nearly identical across different samples, severely limiting exploration. As a result, continuing RL training with parallel sampling no longer yielded performance improvements—effectively, the model had stagnated with no gradients left to optimize. This behavior highlights the fundamental flaw in parallel sampling: when the diversity is lost, further training is ineffective because the model fails to explore new strategies.

## 5.1   SEQUENTIAL SAMPLING CAN SAVE DEAD POLICY

As shown in Figure 6, in the Countdown task, the RAGEN agent with entropy regularization quickly improves its success rate during the first 100 steps, but from step 100 to step 200, learning nearly stagnates. During this phase, we observe that the Pass@16/Pass@1 ratio declines towards 1, indicating that the model's outputs are collapsing and becoming nearly identical, which signifies the onset of a "dead policy" where further training leads to little to no progress.

To combat this, we resumed training at this checkpoint using sequential sampling. After continuing training with the sequential sampling approach, we observed a notable recovery. The Pass@16/Pass@1 ratio increased, indicating a return to greater diversity in the model's outputs. Thus, sequential sampling offers a robust method for reviving exploration when an RL-trained agent stagnates, ensuring sustained improvement and preventing the model from being trapped in a local optimum. By diversifying the search for solutions through sequential candidate generation, the model can break out of the "dead policy" state and continue learning, which is especially important in tasks requiring complex decision-making and exploration.

## 6 RELATED WORK

**Reinforcement Learning for Large Language Models**  Reinforcement learning (RL) has been widely adopted to align large language models (LLMs) with human preferences. Methods such as *Reinforcement Learning from Human Feedback* (RLHF) (Ouyang et al., 2022) use human preferences to shape model behavior through reward modeling and policy optimization algorithms like PPO (Schulman et al., 2017). Beyond alignment, RL has also been applied to enhance the reasoning capabilities of LLMs. *RL with Verifiable Rewards* (RLVR) (Lambert et al., 2025) leverages automated reward signals for tasks with objective correctness criteria. Approaches like DeepSeek-R1-Zero (DeepSeek-AI et al., 2025) demonstrate that large-scale RL training can elicit advanced reasoning without supervised fine-tuning.

**Exploration Problem in RL Training**  Recent work points out that for most RL algorithms(e.g. PPO, GRPO), the exploration ability is limited by the corresponding base model (Yue et al., 2025), hindering the continuous scaling of model performance. A key limitation of RL in LLMs is the tendency toward *over-exploitation*, leading to reduced diversity and *entropy collapse* (Cheng et al., 2025; Cui et al., 2025). To mitigate this issue, previous work proposed several approaches, including developing GRPO variants (Yu et al., 2025; Liu et al., 2025b), adjust magtitude of based on token-level or sentence-level entropy (Chen et al., 2025b), or reward shaping (Chen et al., 2025c). Despite partially mitigating the issue of entropy collapse, most methods still rely on parallel sampling, where all samples follow the same probability distribution, inherently limiting exploration.

**Parallel Sampling in LLM Reasoning**  Some studies introduce parallel thinking within a single rollout—for example, generating several different ideas in parallel at appropriate stages and then aggregating them—to consider multiple lines of thought simultaneously and speed up generation (Brown et al., 2024; Rodionov et al., 2025). Existing work either uses constructed datasets for SFT or apply RL to optimize this process (Chen et al., 2025a; Pan et al., 2025; Zheng et al., 2025b). However, for faster inference, the parallel idea-generation branches are independent of one another, and thus diversity among ideas is not ensured; as training progresses, the model's outputs can still lose diversity (i.e., undergo policy collapse). Works like Yao et al. (2023) and Zhang et al. (2024) provide more diverse branches. However, they rely heavily on reliable heuristics or external verifiers. When state estimate is inaccurate, it's easy to overlook paths that are actually correct. That's what exploration is really for: *uncovering those trajectories the model wouldn't deem correct on its own*. Our approach directly exposes the model to the diverse strategies produced by sequential sampling, effectively introducing "off-policy" learning so the model can absorb these new signals and learn from them.

## 7 CONCLUSION

We showed that parallel sampling in nowadays RL algorithms structurally limits exploration, leading to entropy collapse and stagnation. By introducing sequential sampling with a two-stage framework—method drafting and guided solution generation—we preserve diversity, expand coverage of valid strategies, and even revive collapsed policies. Experiments on synthetic, agentic, and reasoning tasks confirm that structured sequential exploration improves both diversity and success rates, offering a simple yet effective path toward sustained gains in LLM reasoning.

## REPRODUCIBILITY STATEMENT

We have taken several steps to facilitate independent reproduction of our results. The sequential sampling protocol and training/inference workflow are summarized in Section 2.2, Section 3 and formalized in Algorithm 1. The construction and evaluation protocol for the synthetic *Path Exploration* task are detailed in Appendix A.1, while task descriptions and configurations for the agent benchmarks (Sokoban, Countdown, FrozenLake) are in Appendix A.2. We provide an anonymous, downloadable code repository in the supplementary materials, including training/evaluation scripts, configuration files with fixed random seeds, prompt templates. Together, these pointers cover the implementation, data, and evaluation details needed to reproduce and extend our findings.

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

# A  TASK DESCRIPTION

## A.1  DATA CONSTRUCTION OF PATH EXPLORATION TASK

**Ground-truth treasure locations**  We prompt the model to generate about 50 city-level prefixes (e.g., "Guangdong Province – Shenzhen City – ..."). For each generated prefix, we then prompt the model again to complete the district/county name. Aggregating across all cities yields roughly 200 different candidate locations (normalized to the format "Province – City – District/County"). From these 200 candidates, we sample 20 as the ground-truth set and verify each one against public administrative gazetteers and online maps to ensure it corresponds to a real, properly named locality.

For reproducibility, we use a fixed random seed; duplicates and ambiguous names are deduplicated and standardized prior to verification.

The filtered 20 treasure locations are as below:

> - Anhui Province – Hefei City – Feidong County
> - Heilongjiang Province – Harbin City – Harbin Economic & Technological Development Zone
> - Tianjin Municipality – Tianjin Economic-Technological Development Area
> - Guangdong Province – Shenzhen City – Shenzhen High-tech Zone
> - Sichuan Province – Deyang City – Deyang Economic Development Zone
> - Jiangsu Province – Lianyungang City – Lianyungang Economic & Technological Development Zone
> - Heilongjiang Province – Harbin City – Songbei District
> - Jiangxi Province – Ganzhou City – Ganzhou High-tech Industrial Development Zone
> - Henan Province – Luoyang City – Luolong District
> - Ningxia Hui Autonomous Region – Yinchuan City – Yinchuan Economic & Technological Development Zone
> - Sichuan Province – Chengdu City – Jinniu District
> - Guangdong Province – Guangzhou City – Baiyun District
> - Jiangsu Province – Nanjing City – Xuanwu District
> - Jiangxi Province – Nanchang City – Jinggangshan District
> - Hunan Province – Changsha City – Yuhua District
> - Zhejiang Province – Hangzhou City – Xihu District
> - Hunan Province – Changsha City – Yuelu District
> - Fujian Province – Xiamen City – Siming District
> - Gansu Province – Lanzhou City – Chengguan District
> - Henan Province – Luoyang City – Luolong District

**Model Prompt**

> One day, an alien spaceship passed by and randomly dropped 20 treasures in different districts/counties across China. Where do you think they might have landed? Please output a location from province to district/county. Example: Guangdong Province – Shenzhen City – Nanshan District.

### A.2 AGENTIC TASKS

The tasks used to evaluate the performance of SESA agent include Sokoban, Countdown, and FrozenLake, each commonly employed in reinforcement learning (RL) research to assess an agent's decision-making and exploration capabilities.

- Sokoban: Sokoban is a classic puzzle game where an agent must push boxes into target locations within a maze. The task requires the agent to plan its moves carefully, as each box can only be moved once per step, and there are limited available spaces.
- Countdown: Countdown is a mathematical game in which the agent is provided with a set of numbers and must use basic arithmetic operations to reach a target number.
- FrozenLake: FrozenLake is a classical RL task where an agent navigates a grid representing a frozen lake, with the goal of reaching a target while avoiding hazardous areas (e.g., water). The challenge lies in the agent's ability to explore the environment safely, as certain areas of the lake are dangerous, and the agent must balance exploration with avoiding risks.

These tasks are widely used benchmarks in RL research, designed to evaluate various aspects of agent behavior such as exploration, strategy diversity, and the stability of learned policies.

Table 2: Hyperparameters and training setup.

| Component | Hyperparameter | Value |
|---|---|---|
| **Actor (policy)** | Learning rate | $1.0 \times 10^{-6}$ |
| | Weight decay | 0.01 |
| | LR scheduler | Constant |
| | Entropy coefficient | 0.001 |
| | Use KL in reward | False |
| | Use KL loss | False |
| **DAPO** | Clip low | 0.2 |
| | Clip high | 0.28 |
| **Training setup** | Total training steps | 200 |
| | Global train batch size per update | 32 |
| | Mini-batch size | 32 |
| | Training seed | 12345 |
| | Validation seed | 123 |

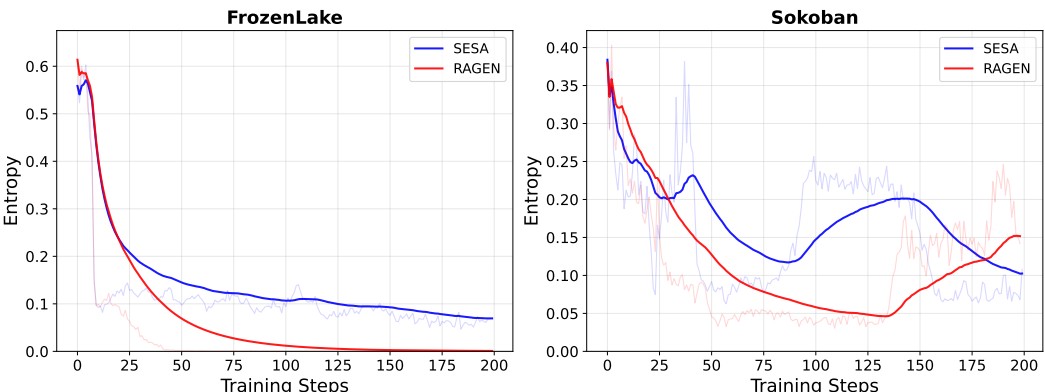

Figure 7: Training-time policy entropy on FrozenLake and Sokoban.

## B  TRAINING DETAILS

### B.1  TRAINING CURVES

#### B.1.1  POLICY ENTROPY

To observe whether SESA actually maintains a higher policy entropy, we report the evolution of entropy throughout RL training under SESA and RAGEN. As shown in Figure 7, solid curves denote the moving average over training steps, and lightly shaded curves show raw per-step values. Across both tasks, parallel sampling exhibits rapid entropy collapse—approaching near-zero entropy on FrozenLake—indicating premature convergence to a narrow set of behaviors. In contrast, SESA relatively maintains higher entropy, demonstrating preserved exploration capacity and preventing the degeneracy that stalls learning in the baseline.

#### B.1.2  IN-GROUP STD

Figure 8 shows the standard deviation of rewards within each batch across training steps for SESA and RAGEN. The standard deviation serves as a proxy for the diversity of strategies explored during training. While the baseline's reward standard deviation approaches zero as training progresses, indicating policy collapse, SESA maintains a significantly higher standard deviation, suggesting ongoing exploration and non-trivial gradient updates. This reinforces the claim that SESA prevents the premature convergence observed in traditional parallel-sampling methods.

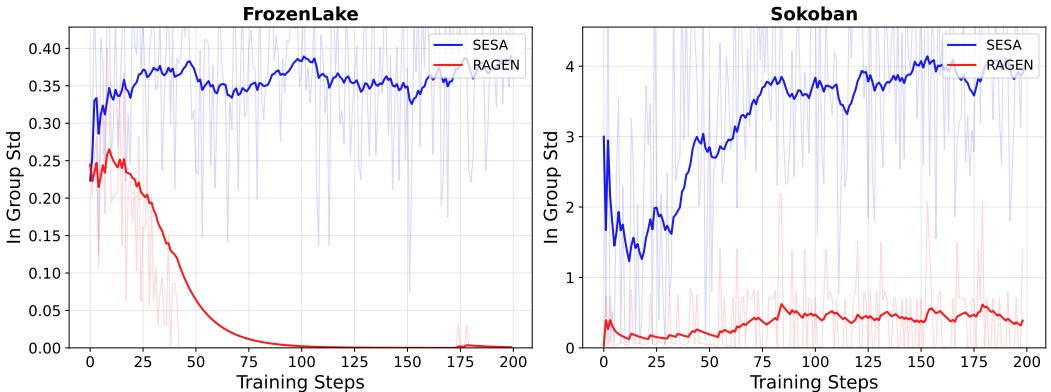

Figure 8: In-group reward standard deviation during training on FrozenLake and Sokoban.

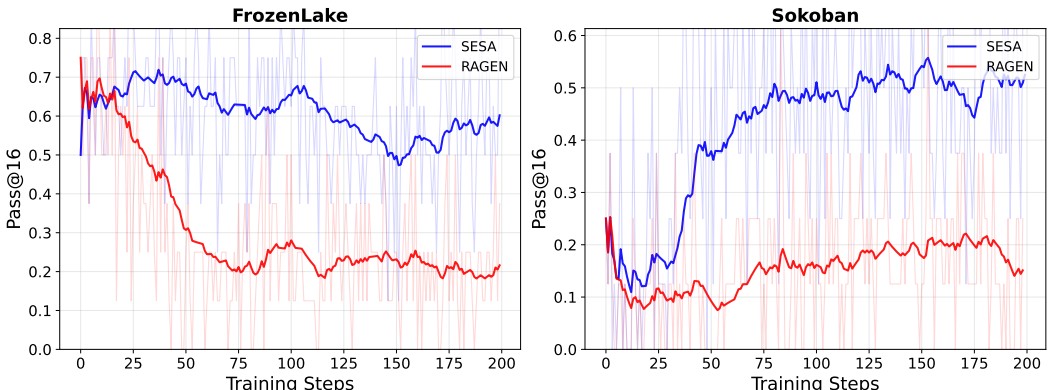

Figure 9: Pass@16 performance during training on FrozenLake and Sokoban.

### B.1.3 PASS@16 METRIC

Figure 9 shows the Pass@16 metric, which measures the proportion of tasks for which at least one successful trajectory is found out of 16 generated samples. The results highlight the improvement in exploration capacity enabled by SESA compared to RAGEN. While the baseline shows a slower increase in Pass@16, indicating limited exploration and policy collapse, SESA consistently outperforms the baseline, demonstrating a broader and more diverse exploration of solution strategies across both tasks.

## C   CASE STUDY: SOKOBAN STRATEGIES DISCOVERED BY SESA

In this case study, we analyze the strategies discovered by the SESA-trained agent on the Sokoban task. A typical strategy observed during training involves the agent sometimes having to:

> "Move from the starting point, pass through the goal, go around to the back of the box, and then push the box onto the target."

Consider the following Sokoban map:

```
######
###P##
###X_#
#___O#
#____#
######
```

In this map, the symbols represent the following:

- # : Wall
- _ : Empty space
- O : Target
- X : Box
- P : Player

To solve this level, the player must execute the following action sequence: *down, right, down, down, left, left, up, right*, which requires the agent to "go past" the goal and then loop around behind the box—a non-trivial strategy that is not immediately obvious from greedy push attempts.

We selected 10 maps where such "go past then wrap around the box" behavior is necessary for success. For these maps:

- The **base model** achieves Pass@16 = 0, meaning it never finds a successful trajectory.
- After 200 training steps, **DAPO+Entropy** still fails to solve any of these levels (Pass@16 = 0).
- In contrast, the **SESA**-trained model achieves Pass@16 = 0.5 on this subset, i.e., it successfully solves 50% of these challenging maps.

Table 3: Performance comparison on the "go past then wrap around the box" maps.

| Method | Pass@16 | Number of actions |
|---|---|---|
| DAPO+Entropy | 0.0 | 5.9 |
| SESA | **0.5** | 8.8 |

Moreover, we observe that the average number of actions taken per map is substantially higher for the SESA-trained policy. This suggests that SESA-trained agents are more willing to try multiple different local strategies before converging on a successful one, instead of committing early and prematurely terminating exploration.

## D   THE USE OF LARGE LANGUAGE MODELS

We used LLMs only for light *text polishing* (clarity and grammar) on certain paragraphs and for *literature discovery* via deep-research. All technical content, experiments, figures, and analyses were authored and verified by the authors.

