# OpenReview forum: "The Road Less Traveled: Enhancing Exploration in LLMs via Sequential Sampling"
_ICLR.cc/2026/Conference — Submitted to ICLR 2026_

### Official Review · Reviewer_5BiQ · 2025-10-27

**Soundness:** 3
**Presentation:** 2
**Contribution:** 3
**Rating:** 4
**Confidence:** 3

**Summary:**

This paper identifies a practical failure mode for RL fine-tuning of LLMs: when multiple rollouts are drawn i.i.d. from the same autoregressive policy (parallel sampling), the policy tends to collapse to a small set of high-reward outputs and exploration vanishes. To counter this, the authors propose SESA, a two-stage sequential sampling framework: (1) sequentially generate short method sketches conditioned on previous sketches to force diversity, and (2) expand each sketch in parallel into full solutions. During training SESA uses sequential sketching to diversify rollouts but computes policy updates with likelihoods conditioned only on the original input. Empirically SESA is evaluated on several reasoning/agent benchmarks. Reported results show large improvements in Pass@k and success rates versus parallel-sampling RL baselines.

**Strengths:**

1. The topic of enhancing exploration in RL finetuning for LLM is important
2. The proposed method appears well-founded and effective, supported by extensive empirical evidence.

**Weaknesses:**

1. **Efficiency concern:** The proposed sequential sketching approach inevitably increases computational cost during rollouts. However, the paper does not provide an analysis of computational efficiency, such as wall-clock time or resource overhead, which would help quantify this trade-off.
2. **Limited model and baseline coverage:** Experiments are conducted only on Qwen-7B, making it unclear whether the observed improvements generalize to other major LLM families. In addition, the set of baselines is limited. It would be valuable to include comparisons with other exploration-oriented methods, such as entropy-based methods[1].
3. **Mismatch between training sampling and evaluation distribution:** The method generates candidate responses conditioned on prior sketches, yet policy updates are computed using likelihoods conditioned solely on the input. Although the paper claims this design maintains training–evaluation consistency, the statistical implications, such as importance weighting, off-policy bias, and variance, are not thoroughly analyzed.
4. **Ablation and sensitivity analysis:** The criteria for selecting sketch lengths and templates are not well justified. Further analysis is needed to understand how sensitive performance is to sketch length, template wording, and other design choices.
5. **Implementation clarity:** The paper lacks sufficient details on the implementation of both the proposed method and baselines, including prompt sketch formats, hyperparameters, and component configurations. Providing these details would improve reproducibility.
6. **Missing metrics:** The paper does not report pass@1 results, which are standard for evaluating reasoning performance. Additionally, other metrics reflecting training dynamics, such as policy entropy, are also important to report.

[1] The entropy mechanism of reinforcement learning for reasoning language models

**Questions:**

see weaknesses

---

> ### Author Response · Authors · 2025-12-03
> **Reply 1/n**
>
> ### **W1：“Efficiency concern”**
> We have now measured the overhead more systematically. Using the Countdown task as a representative example, we compare RAGEN (parallel rollout) and SESA (sequential rollout) at 7B and 3B scale:
>
>
> | **Method** | **Training Time (s/step)** | **Inference Time (s/step)** | **GPU Memory (GB)** |
> | ---------- | -------------------------- | --------------------------- | ------------------- |
> | RAGEN (7B) | 42.2                       | 11.8                        | 71.0                |
> | SESA (7B)  | 48.4                       | 11.1                        | 71.0                |
> | RAGEN (3B) | 25.7                       | 3.3                         | 71.3                |
> | SESA (3B)  | 26.3                       | 3.4                         | 71.6                |
>
>
> - During training, SESA is slightly slower, due to the need to generate sketches before solutions in rollout.
> - During inference, the time per step is very similar because of the same generation process.
> - GPU memory usage is essentially unchanged. With vLLM, a fixed fraction of GPU memory is reserved for generation, so peak memory is dominated by model parameters rather than the modest context-length increase due to sketches.
>
> ----
> ### **W2:“Limited model and baseline coverage”**
>
> We intentionally used smaller models (Qwen2.5-3B-Instruct and 1.5B) for the experiments reported in Table 1. To further examine the effectiveness of SESA at larger scales, we additionally conducted experiments using a 7B model on Sokoban and FrozenLake:
>
> | **Method**     | **Sokoban** | **FrozenLake** |
> | -------------- | ----------- | -------------- |
> | Base Model     | 0.23        | 0.20           |
> | DAPO + Entropy | 0.31        | 0.23           |
> | **SESA**       | **0.46**    | **0.28**       |
>
> And we ran additional experiments incorporating a stronger baseline inspired by "Pass@k Training for Adaptively Balancing Exploration and Exploitation of Large Reasoning Models". Concretely, we introduce a “DAPO + Pass@k reward” variant. On the small-scale agent experiments, we now obtain:
>
> | **Method**           | **Sokoban** | **Countdown** | **FrozenLake** |
> | -------------------- | ----------- | ------------- | -------------- |
> | Base Model           | 0.09        | 0.15          | 0.19           |
> | DAPO                 | 0.16        | 0.50          | 0.21           |
> | DAPO + Entropy       | 0.15        | 0.53          | 0.21           |
> | DAPO + Pass@k reward | 0.24        | 0.52          | 0.24           |
> | **SESA**             | **0.34**    | **0.57**      | **0.26**       |
>
>
> These results show that SESA still provides a substantial further gain on top of these stronger baselines.
>
> Finally, we want to emphasize more clearly that **sequential sampling is orthogonal to many of these techniques**. Our method modifies the **sampling paradigm** (sequential vs. parallel), whereas entropy regularization and Pass@k-style rewards act at the **loss/reward level**. In principle, they can be combined, suggesting that the benefits of sequential sampling are complementary to, and potentially additive with, existing methods.
>
> ----

---

> ### Author Response · Authors · 2025-12-03
> **Reply 2/n**
>
> ### **W3: Mismatch between training sampling and evaluation distribution**
>
> We agree that our method introduces a degree of off-policy training and that we do not perform importance correction. This choice is deliberate.
>
> In SESA, we encourage the model to generate *diverse* answers by conditioning later samples on earlier ones. As a result, the probability of a sequentially sampled response $y$ under the parallel policy, $\pi_{\text{parallel}}(y)$, can be very small.
>
> In the extreme case where the policy has already collapsed, the model assigns all probability mass to a single response $y\^\*$ and $0$ probability to any alternative $y \neq y\^\*$. For such a $y \neq y\^\*$ obtained via sequential sampling, the importance weight would be
>
> $$\frac{\pi_{\text{parallel}}(y)}{\pi_{\text{sequential}}(y)} = 0$$, so importance sampling would completely discard precisely those “unlikely under the collapsed policy” samples that SESA is designed to **recover** and learn from. In other words, applying importance sampling would directly conflict with the main motivation of SESA: to retain and leverage solutions that the baseline (parallel) policy would almost never generate.
>
> We also evaluated an importance-corrected variant (“SESA+IS”) in our experiments. We found that adding importance sampling slightly hurts performance. For example, on Sokoban (Pass@1):
>
> | **Method**   | **Sokoban (Pass@1)** |
> | ------------ | -------------------- |
> | Base Model   | 0.09                 |
> | DAPO+Entropy | 0.16                 |
> | SESA         | **0.34**             |
> | SESA+IS      | 0.30                 |
>
> These results suggest that while the training objective is indeed slightly biased by treating sequentially sampled trajectories as on-policy under $\pi(\cdot \mid x)$, this bias is *beneficial* in practice: it allows the model to absorb valuable off-policy trajectories that would otherwise be suppressed by importance weights near zero.
>
>
> ### **W4: Ablation and sensitivity analysis**
>
> We have run an ablation study on the **number of sequentially generated trajectories per question**. Concretely, in Sokoban we fix:
>
>
>
> - Batch size (number of questions per batch) = 8
> - Total number of trajectories per question = 16
>
>
>
> For a given question, we let each sequential “round” generate $k$ trajectories, and we repeat this $m$ times, so that $k \times m = 16$. The results are:
>
> | **Setting**                          | **Sokoban (Pass@1)** |
> | ------------------------------------ | -------------------- |
> | $k = 1$, $m = 16$ (fully parallel)   | 0.16                 |
> | $k = 2$, $m = 8$                     | 0.30                 |
> | $k = 4$, $m = 4$                     | **0.34**             |
> | $k = 8$, $m = 2$                     | 0.21                 |
> | $k = 16$, $m = 1$ (fully sequential) | 0.13                 |
>
> Our best setting is $k = 4$, $m = 4$: each question generates 4 different trajectories per sequential round, repeated 4 times.
>
>
>
> We can interpret the trend as follows:
>
>
>
> - With very small $k$ (e.g., $k = 1$), the rollout process equals standard parallel sampling and does not leverage sequential diversity.
>
> - With very large $k$ (e.g., $k = 16$), we over-constrain the model to produce highly diverse responses in a single shot. The late trajectories in the sequence are then either:
>
>
>
>   - Very low-probability and often incorrect, or
>
>   - Repetitions / weak variations of earlier ones.
>
>     This leads to poor learning, sometimes even worse than purely parallel sampling.
>
>
>
> - A moderate $k$ (e.g., $k = 4$) strikes a balance: SESA encourages diversity while still allowing a reasonable fraction of high-quality trajectories.
>
>
> Regarding sketch length and prompt structure, we designed sketches to be short, high-level plans (e.g., “first search around the target, then approach the box from behind”) rather than full solutions. Longer sketches increase context length and may cause later solutions to copy earlier ones, so we deliberately keep them concise.

---

> ### Author Response · Authors · 2025-12-03
> **Reply 3/n**
>
> ### **W5: "Implementation clarity"**
>
> We appreciate this comment and will include the key hyperparameters in the revised manuscript. Below are the main settings used in our experiments:
>
> | **Component**      | **Hyperparameter**                 | **Value**            |
> | ------------------ | ---------------------------------- | -------------------- |
> | **Actor (policy)** | Learning rate                      | $1.0 \times 10^{-6}$ |
> |                    | Weight decay                       | 0.01                 |
> |                    | LR scheduler                       | constant             |
> |                    | Entropy coefficient                | 0.001                |
> |                    | Use KL in reward                   | False                |
> |                    | Use KL loss                        | False                |
> | **DAPO**           | Clip low                           | 0.2                  |
> |                    | Clip high                          | 0.28                 |
> | **Training setup** | Total training steps               | 200                  |
> |                    | Global train batch size per update | 32                   |
> |                    | Mini-batch size                    | 32                   |
> |                    | Training seed                      | 12345                |
> |                    | Validation seed                    | 123                  |
>
> ----
>
> ### **W6: "Missing metrics"**
>
> There may be a misunderstanding — in Table 1, the metric we report as **“success rate” is exactly the pass@1 result**. This corresponds to the proportion of prompts for which the model produces a correct solution in a single attempt, which aligns with the standard pass@1 definition for reasoning tasks.
>
> As for entropy, in Appendix B.1.1, we report the entropy of the policy during training on two tasks (FrozenLake, Sokoban). We observe that:
>
> - For the baseline (parallel sampling), the entropy is relatively low and, in the case of FrozenLake, eventually approaches **0** in the later stages of training, indicating severe policy collapse.
> - Under SESA, the entropy remains relatively higher throughout training on both tasks, especially on FrozenLake, where the baseline entropy nearly vanishes.
>
>
>
>
> To further quantify whether training is still effectively happening, we monitor the **standard deviation of rewards within each group**. A low reward standard deviation (std) implies near-zero gradients; we believe this is a useful proxy for both ongoing learning and diversity in strategies. If std = 0, then RL training is effectively stalled.
>
>
>
> We observe that the in-group reward std under SESA is consistently higher than that of the baseline across training. This indicates that SESA maintains non-trivial gradients and avoids degenerating into a single, deterministic behavior pattern.
>
>
>
> These observations show that SESA does more than change prompts at the surface level: it keeps the policy in a high-entropy, exploratory regime, rather than converging prematurely to a narrow set of strategies.

---

### Official Review · Reviewer_8rvh · 2025-10-30

**Soundness:** 2
**Presentation:** 3
**Contribution:** 2
**Rating:** 2
**Confidence:** 4

**Summary:**

This paper introduces SESA (Sequential Sampling), a two-stage reinforcement learning framework aimed at enhancing exploration in large language models (LLMs). The method replaces traditional parallel sampling with sequential sampling—each output is conditioned on previously generated samples to promote diversity and prevent policy collapse. Experiments across synthetic, reasoning, and agentic tasks show modest gains in diversity and success rates, suggesting potential benefits for exploration in RL-based LLM training.

**Strengths:**

•	Clarity and readability: The paper is well written and easy to follow, with clear descriptions of motivation, method, and results.
•	Relevance: Exploration and diversity in RL training for LLMs are timely and important topics, especially as reinforcement fine-tuning becomes central to reasoning model development.
•	Conceptual simplicity: The idea of conditioning subsequent samples on earlier ones is intuitively appealing and could be a useful heuristic to improve diversity during training.

**Weaknesses:**

•	Misleading framing and terminology: Despite being presented as a new “sampling” method, the approach does not involve actual sampling from the model’s stochastic policy but instead relies on modified prompting. This makes the title and framing potentially misleading.
•	Lack of cost and efficiency analysis: The sequential approach introduces longer input contexts (due to inclusion of sketches), yet the computational or latency overhead is not quantified or discussed. This omission is critical for evaluating practicality.
•	Synthetic evaluation questionable: The “Path Exploration” task does not convincingly demonstrate that sequential sampling uncovers meaningful “strategies.” It is unclear what constitutes a “strategy” in this simplified guessing setup, and whether the gains would persist in more realistic scenarios.
•	Unclear experimental setup: Important details such as which LLMs are used in each experiment (e.g., in Table 1) are missing, making reproducibility and cross-model generalization claims difficult to assess.
•	Inflated performance reporting: Percentage improvements are reported relative to the base model rather than the relevant baseline (e.g., RAGEN), leading to overstated claims. Standard deviations and number of runs are not reported, limiting statistical reliability.

**Questions:**

1.	Why is sequential conditioning referred to as “sampling”? Could you clarify how this differs from prompting with additional context rather than modifying the model’s sampling distribution?
2.	Why is no sketch used during evaluation? Would incorporating sketch conditioning at test time further improve results?
3.	The authors mention deliberately focusing on weaker base models. How does this choice affect the generality and relevance of the proposed method for stronger LLMs?
4.	Can you quantify the additional computational cost (context length, inference time, memory) introduced by sequential sampling compared to parallel sampling?
5.	How is the number of sequential samples (G) chosen, and what happens if it becomes too large or too small?

---

> ### Author Response · Authors · 2025-12-03
> **Reply 1/n**
>
> ### **W1 & Q1: Misleading framing and terminology (“sequential sampling”)**
>
> > *“Despite being presented as a new ‘sampling’ method, the approach does not involve actual sampling from the model’s stochastic policy but instead relies on modified prompting.”*
>
>
> We appreciate this point and agree that our terminology can be clearer.
>
> In the current version, we use “sequential sampling” to emphasize that we are **sampling in the space of policies/trajectories**, not at the token level with a different sampler. Concretely, the key difference from parallel sampling is that each response is drawn from a **history-conditioned distribution**:
>
> $$r_i \sim \pi(\cdot \mid q, r_1,\dots,r_{i-1})$$
>
> which is induced by explicitly conditioning on previously generated sketches/solutions in the prompt. This conditioning **does change the effective sampling distribution over full responses**, even though the underlying decoding algorithm remain the same.
>
>
> That said, we agree that the term “sequential sampling” can be misleading in the LLM context, where “sampling” is often understood as purely token-level stochastic decoding under a fixed prompt. A more precise terminology for our procedure is **sequential rollout** under history-dependent prompting.
>
> In the revision, we will use terms like “sequential rollout”, and clarify early in the paper that our contribution is a **rollout/training paradigm** (sequential vs. parallel) rather than a new token-level sampler.
>
>
> ------
>
>
>
> ### **W2 & Q4: Lack of cost and efficiency analysis**
>
>
> > *“The sequential approach introduces longer input contexts (due to inclusion of sketches), yet the computational or latency overhead is not quantified or discussed.”*
>
> We agree that the cost–efficiency trade-off should be made explicit.
>
>
> 1. **Sketch length.** In all experiments, we cap sketches at **100 tokens**. Compared with the overall context length of the tasks we consider, this overhead is small and has a negligible impact on memory and time.
> 2. **Training and inference time.** We have now measured the overhead more systematically. Using the **Countdown** task as a representative example, we compare RAGEN (parallel rollout) and SESA (sequential rollout) at 7B and 3B scale:
>
>
> | **Method** | **Training Time (s/step)** | **Inference Time (s/step)** | **GPU Memory (GB)** |
> | ---------- | -------------------------- | --------------------------- | ------------------- |
> | RAGEN (7B) | 42.2                       | 11.8                        | 71.0                |
> | SESA (7B)  | 48.4                       | 11.1                        | 71.0                |
> | RAGEN (3B) | 25.7                       | 3.3                         | 71.3                |
> | SESA (3B)  | 26.3                       | 3.4                         | 71.6                |
>
>
> - During training, SESA is slightly slower, due to the need to generate sketches before solutions in rollout.
> - During inference, the time per step is very similar because of the same generation process.
> - GPU memory usage is essentially unchanged. With vLLM, a fixed fraction of GPU memory is reserved for generation, so peak memory is dominated by model parameters rather than the modest context-length increase due to sketches.

---

> ### Author Response · Authors · 2025-12-03
> **Reply 2/n**
>
> ### **W3. Synthetic evaluation and the definition of “strategy”**
>
> > *“The ‘Path Exploration’ task does not convincingly demonstrate that sequential sampling uncovers meaningful ‘strategies.’ It is unclear what constitutes a ‘strategy’ in this simplified guessing setup…”*
>
>
> We agree that the term “strategy” is overloaded here and that the Path Exploration task is a **stylized** environment. In the Path Exploration setting, a “strategy” is operationally just a **distinct valid path** in the space of city–district combinations that yields reward. It does **not** capture rich semantic behavior or multi-step reasoning in the same way as the agent or math tasks.
>
>
> We fully agree that this synthetic task alone does not demonstrate the discovery of semantically meaningful strategies. That is why our main experiments focus on more realistic tasks (Sudoku, AIME24, and agent environments), where different solutions correspond to **qualitatively different reasoning paths**, and we observe SESA discovering and retaining **new solution patterns** that the base model and parallel RL baselines rarely produce.
>
> We here conduct a case study on Sokoban to analyze the strategies discovered by SESA. A typical strategy we observe during training is that the agent sometimes must:
>
> > “Move from the starting point, pass through the goal, go around to the back of the box, and then push the box onto the target.”
>
> For example, consider the following map:
>
> ```
> ######
> ###P##
> ###X_#
> #___O#
> #____#
> ######
> (The meaning of each symbol: #: wall, _: empty, O: target, X: box, P: player)
> ```
>
> To solve this level, the player must execute the following action sequence:
>
> > down, right, down, down, left, left, up, right.
>
> This requires “going past” the goal and then looping around behind the box—a non-trivial strategy that is not obvious from greedy push attempts.
>
> We selected 10 maps for which such “go past then wrap around the box” behavior is necessary to succeed. For these maps:
>
> - The **base model** has Pass@16 = 0 (it never finds a successful trajectory).
> - After 200 training steps, **DAPO+Entropy** still fails to solve any of these levels (Pass@16 = 0).
> - In contrast, the **SESA**-trained model achieves Pass@16 = 0.5 on this subset, i.e., it solves 50% of these hard maps.
>
>
>
> | **Method**   | **Pass@16** | **Number of actions**                      |
> | ------------ | ----------- | ------------------------------ |
> | DAPO+Entropy | 0.0         | 5.9        |
> | SESA         | **0.5**     | 8.8 |
>
> Moreover, we find that the average number of actions taken per map is substantially higher for the SESA-trained policy.
>
>
> This suggests that SESA-trained agents are more willing to **try multiple different local strategies** before converging on a successful one, instead of committing early and prematurely terminating exploration.
>
>
> ------
>
>
> ### **W4. Unclear experimental setup**
>
>
> > *“Important details such as which LLMs are used in each experiment (e.g., in Table 1) are missing…”*
>
>
> We appreciate this comment and will include the key hyperparameters in the revised manuscript. Below are the main settings used in our experiments:
>
> | **Component**      | **Hyperparameter**                 | **Value**            |
> | ------------------ | ---------------------------------- | -------------------- |
> | **Actor (policy)** | Learning rate                      | $1.0 \times 10^{-6}$ |
> |                    | Weight decay                       | 0.01                 |
> |                    | LR scheduler                       | constant             |
> |                    | Entropy coefficient                | 0.001                |
> |                    | Use KL in reward                   | False                |
> |                    | Use KL loss                        | False                |
> | **DAPO**           | Clip low                           | 0.2                  |
> |                    | Clip high                          | 0.28                 |
> | **Training setup** | Total training steps               | 200                  |
> |                    | Global train batch size per update | 32                   |
> |                    | Mini-batch size                    | 32                   |
> |                    | Training seed                      | 12345                |
> |                    | Validation seed                    | 123                  |
>
>
> ------

---

> ### Author Response · Authors · 2025-12-03
> **Reply 3/n**
>
> ### **W5. Inflated performance reporting**
>
> > *“Percentage improvements are reported relative to the base model rather than the relevant baseline”*
>
> We reported percentage improvements relative to the base model because many tasks are quite challenging and the base model’s absolute performance can be low. In such cases, the same absolute gain corresponds to very different relative improvements across tasks, so normalizing by the base model helps compare effect sizes under a common reference point. For example, on Sokoban, RAGEN+Entropy improves over the base model by about 66.7%, whereas SESA improves by 227.8%, a substantially larger gain. In the revision, we will report both absolute performance and percentage improvements, and explicitly distinguish improvements over the base model from improvements over the strongest baseline in each table, making the relative advantage of SESA clearer.
>
> ----
>
> ### **Q2. Why is no sketch used during evaluation?**
>
>
> The reviewer is right that we **do not use sketches at evaluation time**, and this is by design.
>
> During training:
>
> - Rollout is performed in two stages: we first generate a **sketch**, then a **solution** conditioned on the sketch.
> - However, when computing the RL loss, we **ignore the sketch** and evaluate the log-likelihood of the solution under the **un-sketch-conditioned input** $\log \pi_\theta(\text{solution} \mid q)$, rather than $\log \pi_\theta(\text{solution} \mid q, \text{sketch})$.
>
>
> This is for fair comparison at evaluation stage. Baseline methods (DAPO, RAGEN, etc.) do not use sketches at inference time. If we also used sketches at test time, we would introduce a more expensive inference scheme, which might be unfair. And,  in practical scenarios, users only provide the query $q$ without any sketch. Training the model to perform well when conditioned only on $q$ keeps the inference interface simple and directly comparable to existing RL-fine-tuned models.
>
>
> ------
>
>
> ### **Q3. Relevance for stronger LLMs**
>
>
> To check whether SESA also benefits stronger models, we ran additional experiments with a **7B parameter model** on Sokoban and FrozenLake. The results are:
>
> | **Method**     | **Sokoban** | **FrozenLake** |
> | -------------- | ----------- | -------------- |
> | Base Model     | 0.23        | 0.20           |
> | DAPO + Entropy | 0.31        | 0.23           |
> | **SESA**       | **0.46**    | **0.28**       |
>
> SESA continues to provide substantial gains over both the base model and the entropy-regularized baseline at 7B scale, suggesting that the method is not limited to weak models.
>
>
> **Q5. Choice of the number of sequential samples $G$**
>
>
> We have run an ablation study on the **number of sequentially generated trajectories per question**. Concretely, in Sokoban we fix:
>
> - Batch size (number of questions per batch) = 8
> - Total number of trajectories per question = 16
>
> For a given question, we let each sequential “round” generate $k$ trajectories, and we repeat this $m$ times, so that $k \times m = 16$. The results are:
>
> | **Setting**                          | **Sokoban (Pass@1)** |
> | ------------------------------------ | -------------------- |
> | $k = 1$, $m = 16$ (fully parallel)   | 0.16                 |
> | $k = 2$, $m = 8$                     | 0.30                 |
> | $k = 4$, $m = 4$                     | **0.34**             |
> | $k = 8$, $m = 2$                     | 0.21                 |
> | $k = 16$, $m = 1$ (fully sequential) | 0.13                 |
>
> Our best setting is $k = 4$, $m = 4$: each question generates 4 different trajectories per sequential round, repeated 4 times.
>
>
> We can interpret the trend as follows:
>
> - With very small $k$ (e.g., $k = 1$), the rollout process equals standard parallel sampling and does not leverage sequential diversity.
>
> - With very large $k$ (e.g., $k = 16$), we over-constrain the model to produce highly diverse responses in a single shot. The late trajectories in the sequence are then either:
>
>
>   - Very low-probability and often incorrect, or
>
>   - Repetitions / weak variations of earlier ones.
>
>     This leads to poor learning, sometimes even worse than purely parallel sampling.
>
>
> - A moderate $k$ (e.g., $k = 4$) strikes a balance: SESA encourages diversity while still allowing a reasonable fraction of high-quality trajectories.
>
>
> Regarding sketch length and prompt structure, we designed sketches to be short, high-level plans (e.g., “first search around the target, then approach the box from behind”) rather than full solutions. Longer sketches increase context length and may cause later solutions to copy earlier ones, so we deliberately keep them concise.

---

### Official Review · Reviewer_bYrv · 2025-11-01

**Soundness:** 3
**Presentation:** 3
**Contribution:** 1
**Rating:** 2
**Confidence:** 3

**Summary:**

+ Summary & Contributions
	- The authors focus their attention on the well-known issue of entropy collapse and loss of response diversity during the RL-based fine-tuning of LLMs
	- Rather than generating multiple potential outputs independently and in parallel (leaving high risk of duplicate/redundant samples), the authors advocate for a sequential sampling scheme whereby individual responses are obtained one after another, conditioned on the previously-generated responses. Intuitively, such a sequential generation process makes the model privy to early responses and therefore more likely to generate further diverse candidates.
	- Recognizing the complexity of generating responses conditioned on longer and longer histories of earlier responses, the authors propose a more computationally-sensible two-stage approach. The first pass generates a high-level response strategy/method sketch sequentially to ensure diversity before a second pass to generate full responses based on the diverse sketches in parallel.
	- Empirical results are presented demonstrating improved coverage/diversity of responses as well as better pass@k accuracy than two parallel sampling baselines (DAPO & RAGEN).

**Strengths:**

+ Quality
	- Strengths
		- The idea of conditioning on the history of previous responses to preserve diversity is sensible, albeit obvious, way to ensure diversity. Importantly, the authors recognize the computational impracticalities of such conditioning and offer a more practical, scalable alternative through a two-stage, hierarchical RL procedure.
	- Weaknesses
		* Major
			- If I'm understanding correctly, the quantity labeled by the authors as "Coverage@k" is really just accuracy, right? Of exactly $U = 20$ correct answers, how many were successfully recovered by the model. Why not just call this accuracy? More importantly, the design of the PathExploration task doesn't really seem logical as a barometer for good exploration. It seems like the authors synthetically generate 20 needles ("valid string locations") in a massive, China-sized haystack (all possible Province-City-District locations in China). If these 20 special "treasure points" are chosen independently, then there really is no signal to facilitate good exploration. A LLM just needs to get lucky and stumble into these 20 points by guessing through all the others. This seems analogous to a binary tree where one (instead of 20) special leaf node is chosen to have a reward of 1 while all others are zero. If no other node provides signal for where the rewarding leaf node is, then an agent has no recourse but to stumble around and touch everything. Indeed, these kinds of MDPs can be used to prove exponential gaps in sample complexity between RL and imitation learning [2]. Bearing that in mind and after reading through Appendix A.1, I'm surprised and skeptical of the good coverage reported in Figure 3. It seems plausible that perhaps there is some latent LLM prior skewing towards certain points that result in them being generated with such high frequency after so few evaluation steps. More broadly, while this is still positive for sequential vs. parallel sampling, it doesn't seem all that resemblant of the exploration problem faced by LLMs in general, where there is likely latent structure which characterizes more favorable responses.
			- I'm confused about how the authors are able to obtain empirical support for "Conclusion III" (L208) that sequential sampling following the entropy collapse due to traditional parallel sampling is able to recover response diversity. RL fine-tuning occurs in an on-policy manner, whereby policy-gradient updates are performed on prompt-response pairs generated by the current LLM (policy) in order to try and improve response quality. If the collapse has already occurred, this implies that the LLM is producing tokens (and, therefore, responses) with probability 1 or close to it, making sampling of any other response impossible. Could the authors provide more details on how exactly sequential sampling is able to induce policy-gradient updates that revitalize the diversity of the response distribution?
		* Minor
			- To the best of my knowledge, there is no such fundamental principle as "exploration drives continual improvement" in RL. Standard RL problems are often instantiated such that there are well-defined optimal policies for each sequential decision-making problem; once any of these optimal policies have been attained, no further improvement is possible. This is in contrast to the continual/lifelong RL setting, where a lack of convergence does imply there is always more to learn, but I suspect that is not the point the authors were trying to make here. They would likely be better served by simply calling out the classic exploration-exploitation trade-off.
			- It doesn't seem correct to characterize sequential sampling as inducing off-policy learning (475). It would be interesting to hear why the authors believe that to be the case.

+ Clarity
	- Strengths
		- Overall the paper is well-written.
	- Weaknesses
		* Major
			- The authors make a rather odd choice to introduce their method in Section 3 after having already presented preliminary experiments and conclusions. This is strange and doesn't seem to help the structure of the paper. In fact, it would be clearer for the authors to introduce their method and then present all empirical findings.
		* Minor
			- N/A


+ Originality
	- Strengths
		- The authors' instantiation of diversity-preserving response distributions via sequential sampling is, to the best of my knowlege, novel.
	- Weaknesses
		* Major
			- N/A
		* Minor
			- The idea of generating so-called "sketches" which encapsulate the high-level, abstract structure of some behavior is already established in the hierarchical RL literature [1]. One reasonable interpretation of this paper is that the authors have resurrected this idea from deep RL and brought it to bear on RL fine-tuning of LLMs. One novelty

+ Significance
	- Strengths
		- The positive results demonstrated across the experiments provide a sufficiently compelling picture that would likely incentivize readers to experiment with sequential sampling as a mitigation strategy for LLM response diversity collapse.
	- Weaknesses
		* Major
			- Perhaps the biggest issue facing this paper is that the authors are by no means the first to observe that RL fine-tuning can lead to a collapse and drastic loss of response diversity over time. While their proposed sequential sampling scheme seems like one plausible way to address the issue and improve over traditional, collapsing parallel sampling, the authors have failed to evaluate against other proposed, existing approaches for preserving response diversity (for example, the work of [3] although there are likely many other alternatives). Why not compare to any of the other approaches, for instance, cited in the related work section (L457-462)? By failing to evaluate against a stronger set of baselines that encapsulate alternative methodologies for handling the well-known collapse issue, it's unclear how significant sequential sampling is as a resolution.
		* Minor
			- N/A


+ Final Remarks
	- Overall, there are issues on the axes of quality and significance for this paper that aren't negligible. The lack of comparison to alternative methodologies for mitigating response diversity collapse is the main issue that prevents this paper from being ready for publication at this time.



+ References
	1.  Andreas, Jacob, Dan Klein, and Sergey Levine. "Modular multitask reinforcement learning with policy sketches." In International conference on machine learning, pp. 166-175. PMLR, 2017.
	2. Sun, Wen, Arun Venkatraman, Geoffrey J. Gordon, Byron Boots, and J. Andrew Bagnell. "Deeply aggrevated: Differentiable imitation learning for sequential prediction." In International conference on machine learning, pp. 3309-3318. PMLR, 2017.
	3. Veselovsky, Veniamin, Benedikt Stroebl, Gianluca Bencomo, Dilip Arumugam, Lisa Schut, Arvind Narayanan, and Thomas L. Griffiths. "Hindsight Merging: Diverse Data Generation with Language Models." In The 41st Conference on Uncertainty in Artificial Intelligence.

**Weaknesses:**

Please see above.

**Questions:**

Please see above.

---

> ### Author Response · Authors · 2025-12-03
> **Reply 1/n**
>
> ## **Quality**
>
> ### **On “Coverage@k” vs “accuracy”**
>
>
> > *Comment.* “If I’m understanding correctly, the quantity labeled by the authors as Coverage@k is really just accuracy, right? … Why not just call this accuracy?”
>
> **Response.** We agree that $ \text{Coverage@k} $ is closely related to accuracy, but we intentionally use a different name to highlight that our goal in this synthetic task is to measure **how many of the 20 ground-truth strategies are preserved**, rather than how often the model hits *any* one of them.
>
> Formally, if the model generates $k$ candidates and hits $u$ *distinct* treasures out of $U = 20$ total treasures, we define
>
> $$
> \text{Coverage@k} = \frac{u}{U}.
> $$
>
> This differs conceptually from the standard notion of accuracy. For example, if the model repeatedly generates the **same** correct path:
>
> - The distinct-coverage is $u = 1$, so $\text{Coverage@k} = 1/20$.
> - A naive “accuracy” interpretation might misleadingly be read as $1$, because all generated paths are "correct".
>
>
> ------
>
>
> ### **On the design and realism of the Path Exploration task**
>
> > *Comment.* “It seems like the authors synthetically generate 20 needles in a massive haystack… If these 20 special points are chosen independently, then there really is no signal… It seems plausible that perhaps there is some latent LLM prior skewing towards certain points…”
>
>
> **Response.** The reviewer is right that if we were to randomly pick 20 treasures from the full combinatorial space of valid Chinese locations, a model would almost never find them in a small number of samples, because many such locations are effectively out-of-distribution for the model.
>
>
> The sentence in the appendix that reads “treat every prefecture-level (and above) city in China as a prefix” is misleading and we have corrected it. What we actually do is:
>
> 1. **Prompt the model to generate** about 50 city-level prefixes (e.g., “Guangdong Province – Shenzhen City”).
> 2. For **each generated prefix**, we then prompt the model again to complete the **district/county** name.
> 3. We then sample 20 verified treasures from these model-generated candidates.
>
>
> As a result, every ground-truth treasure is **reachable under the model’s own prior**—they’re all locations the model itself can somehow naturally proposes. This is exactly why we can obtain non-trivial coverage in relatively few steps: the prior is not uniform over “China-sized” space, but concentrated on a subset of locations the model is familiar with.
>
>
> > *Comment.* “It doesn’t seem all that resemblant of the exploration problem faced by LLMs in general…”
>
> **Response.** We agree that the Path Exploration task is a **stylized** abstraction and does not capture all aspects of real-world exploration (e.g., dense intermediate signals, hierarchical structure, etc.). Our aim with this synthetic task is simply to isolate a specific phenomenon that RL with parallel sampling tends to collapse to a small subset of these valid solutions, and sequential sampling preserves a larger set of correct responses.
>
> The situation is, some treasures are more likely under the prior than others, and RL is trying to redistribute probability mass among them. This simplified environment lets us directly measure whether the training procedure **retains or discards** distinct valid solutions in a controlled setting.
>
> We fully agree that real RLVR tasks contain richer structure. That is why we also evaluate on Sudoku, AIME24, and agent benchmarks, where the reward landscape and intermediate structure are much closer to real-world reasoning problems.

---

> ### Author Response · Authors · 2025-12-03
> **Reply 2/n**
>
> ### **On Conclusion III: how can sequential sampling recover diversity after collapse?**
>
> > *Comment.* “If the collapse has already occurred… the LLM is producing tokens with probability 1… making sampling of any other response impossible. How exactly can sequential sampling induce policy-gradient updates that revitalize diversity?”
>
> **Response.** We agree with your diagnosis for the parallel sampling case: once the policy for a given question $q$ has effectively collapsed so that
>
> $$
> \pi(r^* \mid q) \approx 1,
> $$
>
> then with standard prompts and i.i.d. sampling, the model will repeatedly output $r^*$, and RL training has no gradient signal to move away from this single mode.
>
>
>
> The key difference with SESA is that **the sampling distribution is changed by the prompt and conditioning context**. Under sequential sampling, the $i$-th response for the same question is generated as
>
> $$
> r_i \sim \pi(\cdot \mid q, r_1, \dots, r_{i-1}),
> $$
>
> where the prompt explicitly instructs the model to **propose a different strategy** from previous ones. Even if $\pi(\cdot \mid q)$ alone is highly peaked on $r^*$, the conditional distribution $\pi(\cdot \mid q, r_1, \dots, r_{i-1})$ under a “please give a new, different method” prompt is not degenerate in practice, and the model can be nudged into producing alternative candidates.
>
> These “forced” alternative candidates (which differ from the collapsed $r^*$) are then evaluated by the verifiable reward. If some of them achieve higher or comparable reward, the policy-gradient update assigns them positive advantage and **pushes probability mass away from the collapsed mode**.
>
> In contrast, parallel sampling never changes the conditioning; every sample is drawn from $\pi(\cdot \mid q)$ with the same prompt, so once that distribution collapses, no new high-reward modes can be discovered.
>
> ------
>
>
> ### **On the RL phrasing “exploration drives continual improvement”**
>
>
> > *Comment.* “To the best of my knowledge, there is no such fundamental principle as ‘exploration drives continual improvement’ in RL… It would be better to call out the classic exploration–exploitation trade-off.”
>
>
> **Response.** We agree with your point. Our intention was precisely to discuss the **exploration–exploitation trade-off**, not to claim a new fundamental principle. More precisely, our argument is:
>
> - For many RLVR settings, the optimal policy is unknown and the model may be far from any optimal strategy.
> - If the policy’s exploration capacity collapses (e.g., entropy goes to zero), the gradient signal vanishes even though the model is still suboptimal, so learning stalls.
>
>
> In the revision, we will replace the phrase “exploration drives continual improvement” with language that explicitly references the exploration–exploitation trade-off, and clarify that our focus is on preventing premature loss of exploration capacity, rather than claiming that RL admits continual improvement.
>
>
>
> ------
>
>
> ### **On calling sequential sampling “off-policy”**
>
> > *Comment.* “It doesn’t seem correct to characterize sequential sampling as inducing off-policy learning.”
>
> **Response.** Thank you for pointing this out. Our intention was to highlight a **mismatch between the behavior policy used for data collection and the policy under which we evaluate log-likelihoods**, which is off-policy in spirit:
>
> - **During rollout**, samples are generated *conditioned on previously produced sketches or responses*, i.e.,
>   $$
>   r_i \sim \pi_\theta(\cdot \mid q, r_1, \dots, r_{i-1}),
>   $$
>
>   so the *behavior distribution* is
>
>   $$
>   b_\theta(r \mid q) \approx \pi_\theta(r \mid q, \text{history}).
>   $$
>
> - **During training**, however, each sampled response is scored only under the **unconditioned input** $q$, i.e., via $\log \pi_\theta(r \mid q)$, as in standard RL fine-tuning.
>
>
> This mismatch means that we are effectively **treating samples drawn from a history-conditioned distribution as if they were drawn from the unconditioned policy**—i.e., treating what are behaviorally off-policy samples as if they were on-policy when computing gradients. That is the sense in which we used the term “off-policy.”

---

> ### Author Response · Authors · 2025-12-03
> **Reply 3/n**
>
> ## **Clarity**
>
> > *Comment.* “It would be clearer to introduce the method and then present all empirical findings.”
>
> **Response.** Our intention with Section 2 (“Motivating Task”) was to first present a **fully sequential** experiment (no two-stage framework) to isolate and visualize the phenomenon of diversity collapse and recovery in the simplest possible setting. The two-stage framework in Section 3 is then introduced as a **practical, scalable instantiation** of sequential sampling for real-world tasks where full sequentialization is too long and expensive.
>
> That said, we agree that the current structure can feel backwards. In the revision, we will:
>
> - Make the distinction between the “purely sequential motivating experiment” and the **formal two-stage SESA framework** much more explicit.
> - Add forward references from Section 2 to Section 3, and potentially adjust the order or headers so that the reader sees more clearly that Section 2 is an exploratory toy setup, whereas Section 3 contains the method we actually use for the main experiments.
>
>
> ------
>
>
> ## **Originality**
>
>
> > *Comment.* “The idea of generating so-called ‘sketches’ … is already established… One reasonable interpretation is that the authors have resurrected this idea from deep RL and brought it to bear on RL fine-tuning of LLMs.”
>
>
> **Response.** We agree and appreciate the connection to hierarchical RL and policy sketches (e.g., Andreas et al., 2017). Our “method sketches” are indeed conceptually similar to high-level plans in hierarchical RL. We will explicitly cite related hierarchical RL works and discuss them in the related work section.
>
> ------
>
>
> ## **Significance**
>
> ### **On baselines and comparison to other diversity-preserving methods**
>
> > *Comment.* “The authors are by no means the first to observe collapse… Why not compare to other approaches such as [3], etc.? … it’s unclear how significant sequential sampling is as a resolution.”
>
>
> **Response.** We fully agree that response diversity collapse is a well-known issue, and we already acknowledge prior observations and mitigation strategies in the Introduction and Related Work sctions.
>
>
>
> Regarding baselines:
>
>
>
> - Our **base RL baseline** is DAPO, which already incorporates techniques such as clip higher to encourage higher entropy during training.
> - For agent tasks, we additionally compare against **RAGEN** and **RAGEN + entropy regularization**, which is a more explicit entropy-based method.
>
>
> Following the reviewer's suggestion, we ran additional experiments incorporating a stronger baseline inspired by "Pass@k Training for Adaptively Balancing Exploration and Exploitation of Large Reasoning Models". Concretely, we introduce a “DAPO + Pass@k reward” variant. On the small-scale agent experiments, we now obtain:
>
> | **Method**           | **Sokoban** | **Countdown** | **FrozenLake** |
> | -------------------- | ----------- | ------------- | -------------- |
> | Base Model           | 0.09        | 0.15          | 0.19           |
> | DAPO                 | 0.16        | 0.50          | 0.21           |
> | DAPO + Entropy       | 0.15        | 0.53          | 0.21           |
> | DAPO + Pass@k reward | 0.24        | 0.52          | 0.24           |
> | **SESA**             | **0.34**    | **0.57**      | **0.26**       |
>
> We also evaluated a larger model (Qwen2.5-7B-Instruct) on Sokoban and FrozenLake:
>
> | **Method**     | **Sokoban** | **FrozenLake** |
> | -------------- | ----------- | -------------- |
> | Base Model     | 0.23        | 0.20           |
> | DAPO + Entropy | 0.31        | 0.23           |
> | **SESA**       | **0.46**    | **0.28**       |
>
> These results show that SESA still provides a substantial further gain on top of these stronger baselines.
>
>
> Finally, we want to emphasize more clearly that **sequential sampling is orthogonal to many of these techniques**. Our method modifies the **sampling paradigm** (sequential vs. parallel), whereas entropy regularization and Pass@k-style rewards act at the **loss/reward level**. In principle, they can be combined, suggesting that the benefits of sequential sampling are complementary to, and potentially additive with, existing methods.

---

### Official Review · Reviewer_zz1H · 2025-11-01

**Soundness:** 2
**Presentation:** 3
**Contribution:** 2
**Rating:** 4
**Confidence:** 4

**Summary:**

This paper proposes a framework to encourage exploration in reinforcement learning with large language models. Instead of generating samples in parallel, the authors propose to produce a sequence of sketches with a single prompt. The sketches are then expanded into full reasoning paths. The approach aims to maintain diversity and mitigate policy collapse during post-training. Experiments on reasoning datasets (Sudoku, AIME24), and agentic benchmarks (Sokoban, Countdown, FrozenLake) show  performance improvements over parallel-sampling baselines.

**Strengths:**

- The paper tackles a timely problem in reinforcement learning for large language models.
- The paper has a clear motivation, is well written and easy to follow. The idea of self-generated strategic sketches as explorative guidance is novel, interesting and intuitively appealing.
- The proposed framework is lightweight and seems easy to integrate into existing pipelines.
- Empirical results show consistent improvements over RAGEN in toy environments.

**Weaknesses:**

- During rollout, samples are generated conditioned on previously produced sketches, yet during training, each sample’s likelihood is evaluated only under the unconditioned input. This effectively treats off-policy samples as if they were on-policy, introducing bias without importance correction.
- The method is only a prompt-level adaptation. The performance boost comes from sampling with different prompts. It remains unclear whether the approach genuinely maintains exploration entropy or converges to a set of strategies that just yield broader initial coverage due to the prompt design.
- The paper provides no details about the strategies discovered or how they evolve over training.
- There is no ablation study on the effect of the number of sketches, sketch length, or prompt structure on performance.
- Critical hyperparameters (learning rates, clipping thresholds, batch sizes, etc.) are omitted in the manuscript.
- The chosen benchmarks are relatively simple; validation on more complex, real-world environments (e.g., WebArena or WebShop) would strengthen the empirical claims.

**Questions:**

- Would offline training methods (e.g., DPO) not be better suited in this setting?
- Could you report Pass@1, Pass@8, Pass@16, and Pass@64 metrics for AIME24 to better understand diversity scaling?
- Please justify the Pass@16/Pass@1 ratio as a diversity indicator. When two models have equal Pass@16 but different Pass@1, this metric penalizes stronger models.
- Could you provide examples or statistics of the strategies found and how their diversity evolves over training?
- How does the number of sketches affect performance, and how is the optimal number chosen?
- How does model capacity (e.g., smaller vs. larger base LLMs) influence the effectiveness of sequential sampling?

---

> ### Author Response · Authors · 2025-12-03
> **Reply 1/n**
>
> We thank the reviewer for the thoughtful and detailed feedback. Below we respond point-by-point to the identified weaknesses and questions.
>
> ### **W1: “Treats off-policy samples as if they were on-policy”**
>
> We agree that our method introduces a degree of off-policy training and that we do not perform importance correction. This choice is deliberate.
>
> In SESA, we encourage the model to generate *diverse* answers by conditioning later samples on earlier ones. As a result, the probability of a sequentially sampled response $y$ under the parallel policy, $\pi_{\text{parallel}}(y)$, can be very small.
>
> In the extreme case where the policy has already collapsed, the model assigns all probability mass to a single response $y\^\*$ and $0$ probability to any alternative $y \neq y\^\*$. For such a $y \neq y\^\*$ obtained via sequential sampling, the importance weight would be
>
> $$\frac{\pi_{\text{parallel}}(y)}{\pi_{\text{sequential}}(y)} = 0$$, so importance sampling would completely discard precisely those “unlikely under the collapsed policy” samples that SESA is designed to **recover** and learn from. In other words, applying importance sampling would directly conflict with the main motivation of SESA: to retain and leverage solutions that the baseline (parallel) policy would almost never generate.
>
> We also evaluated an importance-corrected variant (“SESA+IS”) in our experiments. We found that adding importance sampling slightly hurts performance. For example, on Sokoban (Pass@1):
>
> | **Method**   | **Sokoban (Pass@1)** |
> | ------------ | -------------------- |
> | Base Model   | 0.09                 |
> | DAPO+Entropy | 0.16                 |
> | SESA         | **0.34**                 |
> | SESA+IS      | 0.30                 |
>
> These results suggest that while the training objective is indeed slightly biased by treating sequentially sampled trajectories as on-policy under $\pi(\cdot \mid x)$, this bias is *beneficial* in practice: it allows the model to absorb valuable off-policy trajectories that would otherwise be suppressed by importance weights near zero.
>
> ### **W2: “Only a prompt-level adaptation” / W2 & Q4: “Does it truly maintain exploration entropy?”**
>
>
>
> The reviewer notes that SESA appears to be “only a prompt-level adaptation” and questions whether it actually maintains exploration entropy rather than simply encoding broader initial coverage in the prompt.
>
>
>
> In Appendix B.1.1, we report the entropy of the policy during training on two tasks (FrozenLake, Sokoban). We observe that:
>
> - For the baseline (parallel sampling), the entropy is relatively low and, in the case of FrozenLake, eventually approaches **0** in the later stages of training, indicating severe policy collapse.
> - Under SESA, the entropy remains relatively higher throughout training on both tasks, especially on FrozenLake, where the baseline entropy nearly vanishes.
>
>
>
>
>
> To further quantify whether training is still effectively happening, we monitor the **standard deviation of rewards within each group** in Appendix B.1.2. A low reward standard deviation (std) implies near-zero gradients; we believe this is a useful proxy for both ongoing learning and diversity in strategies. If std = 0, then RL training is effectively stalled.
>
>
> We observe that the in-group reward std under SESA is consistently higher than that of the baseline across training. This indicates that SESA maintains non-trivial gradients and avoids degenerating into a single, deterministic behavior pattern.
>
>
> These observations show that SESA does more than change prompts at the surface level: it keeps the policy in a high-entropy, exploratory regime, rather than converging prematurely to a narrow set of strategies.

---

> ### Author Response · Authors · 2025-12-03
> **Reply 2/n**
>
> ### **W3 & Q4: “No details about discovered strategies or their evolution”**
>
>
> We here conduct a case study on Sokoban to analyze the strategies discovered by SESA. A typical strategy we observe during training is that the agent sometimes must:
>
> > “Move from the starting point, pass through the goal, go around to the back of the box, and then push the box onto the target.”
>
> For example, consider the following map:
>
> ```
> ######
> ###P##
> ###X_#
> #___O#
> #____#
> ######
> (The meaning of each symbol: #: wall, _: empty, O: target, X: box, P: player)
> ```
>
> To solve this level, the player must execute the following action sequence:
>
> > down, right, down, down, left, left, up, right.
>
> This requires “going past” the goal and then looping around behind the box—a non-trivial strategy that is not obvious from greedy push attempts.
>
> We selected 10 maps for which such “go past then wrap around the box” behavior is necessary to succeed. For these maps:
>
> - The **base model** has Pass@16 = 0 (it never finds a successful trajectory).
> - After 200 training steps, **DAPO+Entropy** still fails to solve any of these levels (Pass@16 = 0).
> - In contrast, the **SESA**-trained model achieves Pass@16 = 0.5 on this subset, i.e., it solves 50% of these hard maps.
>
>
>
> | **Method**   | **Pass@16** | **Number of actions**                      |
> | ------------ | ----------- | ------------------------------ |
> | DAPO+Entropy | 0.0         | 5.9        |
> | SESA         | **0.5**     | 8.8 |
>
> Moreover, we find that the average number of actions taken per map is substantially higher for the SESA-trained policy.
>
>
> This suggests that SESA-trained agents are more willing to **try multiple different local strategies** before converging on a successful one, instead of committing early and prematurely terminating exploration.
>
>
>
> ------
>
>
>
>
>
> ### **W4 & Q5: Effect of the number of sketches and how the optimal number is chosen**
>
>
>
> We have run an ablation study on the **number of sequentially generated trajectories per question**. Concretely, in Sokoban we fix:
>
>
>
> - Batch size (number of questions per batch) = 8
> - Total number of trajectories per question = 16
>
>
>
> For a given question, we let each sequential “round” generate $k$ trajectories, and we repeat this $m$ times, so that $k \times m = 16$. The results are:
>
> | **Setting**                          | **Sokoban (Pass@1)** |
> | ------------------------------------ | -------------------- |
> | $k = 1$, $m = 16$ (fully parallel)   | 0.16                 |
> | $k = 2$, $m = 8$                     | 0.30                 |
> | $k = 4$, $m = 4$                     | **0.34**             |
> | $k = 8$, $m = 2$                     | 0.21                 |
> | $k = 16$, $m = 1$ (fully sequential) | 0.13                 |
>
> Our best setting is $k = 4$, $m = 4$: each question generates 4 different trajectories per sequential round, repeated 4 times.
>
>
>
> We can interpret the trend as follows:
>
>
>
> - With very small $k$ (e.g., $k = 1$), the rollout process equals standard parallel sampling and does not leverage sequential diversity.
>
> - With very large $k$ (e.g., $k = 16$), we over-constrain the model to produce highly diverse responses in a single shot. The late trajectories in the sequence are then either:
>
>
>
>   - Very low-probability and often incorrect, or
>
>   - Repetitions / weak variations of earlier ones.
>
>     This leads to poor learning, sometimes even worse than purely parallel sampling.
>
>
>
> - A moderate $k$ (e.g., $k = 4$) strikes a balance: SESA encourages diversity while still allowing a reasonable fraction of high-quality trajectories.
>
>
>
>
>
> Regarding sketch length and prompt structure, we designed sketches to be short, high-level plans (e.g., “first search around the target, then approach the box from behind”) rather than full solutions. Longer sketches increase context length and may cause later solutions to copy earlier ones, so we deliberately keep them concise.

---

> ### Author Response · Authors · 2025-12-03
> **Reply 3/n**
>
> ### **W5: “Critical hyperparameters omitted”**
>
>
> We appreciate this comment and will include the key hyperparameters in the revised manuscript. Below are the main settings used in our experiments:
>
> | **Component**      | **Hyperparameter**                 | **Value**            |
> | ------------------ | ---------------------------------- | -------------------- |
> | **Actor (policy)** | Learning rate                      | $1.0 \times 10^{-6}$ |
> |                    | Weight decay                       | 0.01                 |
> |                    | LR scheduler                       | constant             |
> |                    | Entropy coefficient                | 0.001                |
> |                    | Use KL in reward                   | False                |
> |                    | Use KL loss                        | False                |
> | **DAPO**           | Clip low                           | 0.2                  |
> |                    | Clip high                          | 0.28                 |
> | **Training setup** | Total training steps               | 200                  |
> |                    | Global train batch size per update | 32                   |
> |                    | Mini-batch size                    | 32                   |
> |                    | Training seed                      | 12345                |
> |                    | Validation seed                    | 123                  |
>
>
> ------
>
> ### **Q1: “Would offline methods (e.g., DPO) be better suited?”**
>
>
> Offline methods like DPO, similar to SFT, require either **ground-truth responses** or a **large number of high-quality preference pairs** to train on. In our setting (Sokoban, Countdown, FrozenLake), such data is not readily available: most trajectories produced by the model are low-reward or invalid, and constructing reliable human preference pairs over long action sequences or reasoning chains would be expensive. By contrast, our environments provide **verifiable rewards**, so the agent can learn directly from interaction. SESA is designed precisely for this online RL regime: it modifies the rollout procedure to maintain exploration and prevent entropy collapse, rather than relying on a fixed offline dataset. When rich labeled or preference data exists, DPO can help initialize a strong policy, while SESA addresses the *online* exploration problem that remains even after such initialization.
>
>
> ------
>
>
> ### **Q3: “Justifying Pass@16/Pass@1 as a diversity indicator”**
>
> We agree with the reviewer that the ratio $\text{Pass@16}/\text{Pass@1}$ can in principle penalize stronger models when two models have the same Pass@16 but different Pass@1.
>
> We use this ratio **not** as a standalone measure of absolute quality, but as an indicator of **how much additional benefit** can be  obtained from multiple rollouts. A higher ratio means that additional samples beyond the first increase success significantly, indicating **diverse** outputs; a ratio close to 1 indicates that sampling more trajectories provides little additional benefit—consistent with **collapse** to a narrow set of outputs.
>
> In our evaluation in the paper, **absolute performance** is evaluated using success rate (Pass@1) on the validation set (Table 1), where SESA consistently improves over the base model and parallel RL baselines. The **$\text{Pass@16}/\text{Pass@1}$ ratio** is used primarily to diagnose **diversity and exploration** in training.
>
>
> To rule out the failure mode described by the reviewer, we also directly compare **Pass@16** of SESA versus RAGEN+entropy (baseline) during training in Appendix B.1.3. We find that SESA achieves higher **absolute** Pass@16, especially in the later stages of training. This confirms that SESA is not merely trading off single-sample performance for diversity.
>
> -----
>
> ### **Q6: "How does model capacity (e.g., smaller vs. larger base LLMs) influence the effectiveness of sequential sampling?"**
>
> We have evaluated SESA on both smaller (3B) and larger (7B) base models and observe consistent benefits across capacities. For the 7B model on Sokoban and FrozenLake, SESA still brings clear gains over both the base model and DAPO+Entropy:
>
> | **Method**   | **Sokoban** | **FrozenLake** |
> | ------------ | ----------- | -------------- |
> | Base Model   | 0.23        | 0.20           |
> | DAPO+Entropy | 0.31        | 0.23           |
> | SESA         | **0.46**    | **0.28**       |
>
> (Results are Pass@1.) The 3B results are reported in Table 1 of the paper and show the same results. Taken together, these experiments indicate that sequential sampling is effective for both smaller and larger LLMs, and its benefits do not rely on a particular model scale.

---

### Meta-Review · Area_Chair_UeNx · 2026-01-05

**Summary:**

The authors present an important problem, that of entropy collapse when optimizing LLMs with RLVR. The presented method is intuitive: sequentially sample candidates, taking into account previous ideas (referred to as sketches). The results show marginal improvements on a variety of tasks such as Sudoku and Frozen Lake. Only in Sudoku were the improvements considerable.

Overall, the reviewers agree with the authors that the problem of entropy is an important one. However, the main disagreement comes with whether the presented results contain enough information to draw conclusions. In particular, a request for stronger baselines are only been partially answered by the authors. Additionally, questions were raised with regards to the ratio of Pass @ 16 / Pass @ 1 as a measure of diversity: indeed it is hard to be convinced that this measure captures it meaningfully. Finally, concerns about the way results were presented were shared, and although it is sometimes a practice in the community, it's likely not something we should encourage.

A concern about off-policyness was raised, which is fair one. However, off-policy is very hard to deal with in practice. Additionally, there was a concern about the training/eval mismatch. This could be an good opportunity, rather than a weakness, for the authros to perform an experiment where the evaluation setup would correspond to a sequential rollout of solutions. This could have strong relevance in many cases where an LLM has to improve on previous attempts, therefore extending the scope of the work.

**Reviewer Concerns:**

Reviewer zz1H: Concerns about the lack of hyperparemeters details would be resolved. Those about off-policyness, on the measure of diversity and model capacity would remain.
Reviewer bYrv: Concerns about metrics would be resolved, however, those about off-policyness and additional baselines would remain.
Reviewer 8rvh: Concerns about training details and the usage of sketches during evaluation would be resolved. Concerns about the way performance is reported would remain.
Reviewer 5BiQ: Concerns about training details, sensitivity analysis and efficiency would be resolved. Those about limited model and baseline coverage would remain.

**Reviewer Scores:**

Reviewer zz1H: it is unlikely the score would change.
Reviewer bYrv: it is unlikely the score would change.
Reviewer 8rvh:  it is possible the score would change.
Reviewer 5BiQ: it is unlikely the score would change.

---

### Decision · Program_Chairs · 2026-01-26

Reject